# DEUP: Direct Epistemic Uncertainty Prediction

**Salem Lahlou**[*]                                                                    *lahlosal@mila.quebec*
*Mila - Quebec AI Institute, Université de Montréal*

**Moksh Jain**[*]                                                                    *moksh.jain@mila.quebec*
*Mila - Quebec AI Institute, Université de Montréal*

**Hadi Nekoei**                                                                    *nekoeihe@mila.quebec*
*Mila - Quebec AI Institute, Université de Montréal*

**Victor Ion Butoi**[†]                                                                    *vbutoi@mit.edu*
*Massachusetts Institute of Technology*

**Paul Bertin**                                                                    *bertinpa@mila.quebec*
*Mila - Quebec AI Institute, Université de Montréal*

**Jarrid Rector-Brooks**                                                                    *jarrid.rector-brooks@mila.quebec*
*Mila - Quebec AI Institute, Université de Montréal*

**Maksym Korablyov**                                                                    *korablym@mila.quebec*
*Mila - Quebec AI Institute, Université de Montréal*

**Yoshua Bengio**                                                                    *yoshua.bengio@mila.queebec*
*Mila - Quebec AI Institute, Université de Montréal*
*CIFAR Fellow*

**Reviewed on OpenReview:** *https://openreview.net/forum?id=eGLdVRvvfQ*

## Abstract

Epistemic Uncertainty is a measure of the lack of knowledge of a learner which diminishes with more evidence. While existing work focuses on using the variance of the Bayesian posterior due to parameter uncertainty as a measure of epistemic uncertainty, we argue that this does not capture the part of lack of knowledge induced by model misspecification. We discuss how the excess risk, which is the gap between the generalization error of a predictor and the Bayes predictor, is a sound measure of epistemic uncertainty which captures the effect of model misspecification. We thus propose a principled framework for directly estimating the excess risk by learning a secondary predictor for the generalization error and subtracting an estimate of aleatoric uncertainty, i.e., intrinsic unpredictability. We discuss the merits of this novel measure of epistemic uncertainty, and highlight how it differs from variance-based measures of epistemic uncertainty and addresses its major pitfall. Our framework, Direct Epistemic Uncertainty Prediction (DEUP) is particularly interesting in interactive learning environments, where the learner is allowed to acquire novel examples in each round. Through a wide set of experiments, we illustrate how existing methods in sequential model optimization can be improved with epistemic uncertainty estimates from DEUP, and how DEUP can be used to drive exploration in reinforcement learning. We also evaluate the quality of uncertainty estimates from DEUP for probabilistic image classification and predicting synergies of drug combinations.

---

[*]Equal Contribution
[†]Work done during an internship at Mila.

# 1 Introduction

A remaining great challenge in machine learning research is purposeful knowledge-seeking by learning agents, which can benefit from estimation of **epistemic uncertainty** (EU), i.e., a measure of lack of knowledge that an active learner should minimize. Unlike **aleatoric uncertainty**, which refers to an intrinsic notion of randomness and is irreducible by nature, epistemic uncertainty can potentially be reduced with additional information (Kiureghian & Ditlevsen, 2009; Hüllermeier & Waegeman, 2019).

In machine learning, EU estimation is already a key ingredient in interactive decision making settings such as active learning (Aggarwal et al., 2014; Gal et al., 2017), sequential model optimization (SMO, Frazier (2018); Garnett (2022)) as well as exploration in reinforcement learning (RL, Kocsis & Szepesvári (2006); Tang et al. (2017)), as EU estimators can inform the learner about the potential information gain from collecting data around a particular area of state-space or input data space.

**What is *uncertainty* and how to quantify it?** In the subjective interpretation of the concept of Probability (De Morgan, 1847; Ramsey, 1926), probabilities are representations of an agent's *subjective* degree of confidence (Hájek, 2019). It is thus unsurprising that probabilities have been widely used to represent uncertainty. Yet, there is not a generally agreed upon definition of uncertainty, let alone a way of quantifying it as a number, as confirmed by a meta-analysis performed by Zidek & Van Eeden (2003). In supervised learning, Bayesian methods, which aim at predicting a conditional probability distribution on the variable to predict, called the *posterior predictive*, are natural candidates for providing uncertainty estimates. Bayesian learning is however more computationally expensive than its frequentist counterpart, and generally relies on approximations such as MCMC (Brooks et al., 2011) and Variational Inference (Blei et al., 2017) methods. Prominent uncertainty estimation methods, such as MC-Dropout (Gal & Ghahramani, 2016) and Deep Ensembles (Lakshminarayanan et al., 2017), both of which are approximate Bayesian methods (Hoffmann & Elster, 2021), use the posterior predictive variance as a measure of uncertainty: if multiple neural nets that are all compatible with the data make different predictions at $x$, the discrepancy between these predictions is a strong indicator of epistemic uncertainty at $x$.

**Pitfalls of using the Bayesian posterior to estimate uncertainty:** Epistemic uncertainty in a predictive model, seen as a measure of lack of knowledge, consists of *approximation uncertainty* - due to the finite size of the training dataset, and *model uncertainty* - due to model misspecification (Hüllermeier & Waegeman, 2019), also called bias. The latter is not accounted for when using variance or entropy as a measure of EU, but has been shown to be important for generalization (Masegosa, 2020) as well as within interactive learning settings like optimal design (Zhou et al., 2003). Approximate Bayesian methods that are widely used in machine learning often suffer from model misspecification, for instance due to the implicit bias induced by SGD (Kale et al., 2021) and the finite computational time. Fig. 1 illustrates with a Gaussian Process (GP) the suboptimal behavior of the GP to estimate EU under model misspecification. The confidence intervals provided by the GP are underestimated when the model is misspecified, which can lead to confidently wrong decisions and poor decisions (see Fig. 1, bottom left).

As a **first contribution**, we systematically analyze the sources of uncertainty and misspecification, and analyze the pitfalls of using discrepancy-based measures of EU (such as variance of the Bayesian posterior predictive), given that they miss out on model uncertainty, which we define as a component of the excess risk, i.e. the gap between the risk (or out-of-sample loss) of the predictor at a point $x$ and that of the Bayes predictor at $x$ (the one with the lowest expected loss, that no amount of additional data could reduce).

As a **second contribution**, we take a step back by considering the fundamental notion of epistemic uncertainty as lack of knowledge, and based on this, we propose to estimate the excess risk as a measure of EU. This leads us to our proposed framework **DEUP**, for **Direct Epistemic Uncertainty Prediction**, where we train a secondary model, called the error predictor, with an appropriate objective and appropriate data, to estimate the point-wise generalization error (the risk), and then subtract an estimate of aleatoric uncertainty if available, or provide an upper bound on EU otherwise. Fig. 1 illustrates in a toy task in which held-out is available, that the epistemic uncertainty estimated by DEUP is more useful to an interactive

learner than those provided by a misspecified GP. It is important to note that, in these settings of interest, DEUP does not use any additional held-out data.

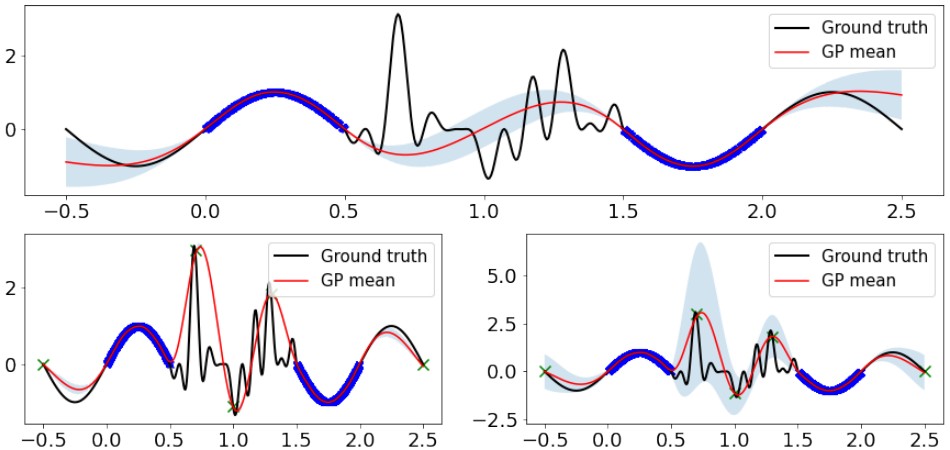

Figure 1: *Top.* A GP is fit on (dark blue) points in $[0, 0.5] \cup [1.5, 2]$. The shaded area represents the GP standard deviation, often used as a measure of epistemic uncertainty: note how it fails badly to enclose the ground function in $[0.5, 1.5]$. *Bottom left.* A second GP fit on the same points, with 5 extra points (green crosses). This second GP predicts almost 0 standard deviation everywhere, even though the first GP significantly underestimated the uncertainty in $[0.5, 1.5]$, because no signal is given to the learner that the region $[0.5, 1.5]$ should be explored more. *Bottom right.* DEUP learns to map the underestimated variances of the first GP to the L2 errors made by that GP, and yields more reasonable uncertainty estimates that should inform the learner of what area to explore.

Accounting for bias when measuring EU is particularly useful to an interactive learner whose effective capacity depends on the training data, for instance a neural network, as it can reduced with additional training data, especially in regions of the input space we care about, i.e. where EU is large. DEUP is agnostic to the type of predictors used, and in interactive settings, it is agnostic to the particular search method still needed to select points with high EU in order to propose new candidates for active learning (Aggarwal et al., 2014; Nguyen et al., 2019; Bengio et al., 2021), SMO (Kushner, 1964; Jones et al., 1998; Snoek et al., 2012) or exploration in RL (Kocsis & Szepesvári, 2006; Osband et al., 2016; Janz et al., 2019).

A unique advantage of DEUP, compared with discrepancy-based measures of EU, is that it can be explicitly trained to care about, and calibrate for estimating the uncertainty for examples which may come from a distribution slightly different from the distribution of most of the training examples. Such distribution shifts (referred to as *feedback covariate shift* in Fannjiang et al. (2022)) arise naturally in interactive contexts such as RL, because the learner explores new areas of the input space. In these non-stationary settings, we typically want to retrain the main predictor as we acquire new training data, not just because more training data is generally better but also to better track the changing distribution and generalize to yet unseen but upcoming out-of-distribution (OOD) inputs. This setting makes it challenging to train the main predictor but it also entails a second non-stationarity for *the training data seen by the error predictor*: a large error initially made at a point $x$ before $x$ is incorporated in the training set (along with an outcome $y$) will typically be greatly reduced after updating the main predictor with $(x, y)$. To cope with this non-stationarity in the targets of the error predictor, we propose, as a **third contribution**, to use additional features as input to the error predictor, that are informative of both the input point *and the dataset* used to obtain the current predictor. We mainly use density estimates and model variance estimates, that sometimes come at no additional cost.

The remainder of the paper is divided as follows:

- In Sec. 2, we describe a theoretical framework for analysing sources of uncertainty and describe a pitfall of discrepancy-based measures of epistemic uncertainty.
- In Sec. 3, we present DEUP, a principled model-agnostic framework for estimating epistemic uncertainty, and describe how it can be applied in a fixed dataset as well as interactive learning settings, and propose means to address the non-stationarities arising in the latter.
- In Sec. 4, we discuss related work on uncertainty estimation as well as error predictors in machine learning.
- In Sec. 5, we experimentally validate that EU estimates from DEUP can improve upon existing SMO methods, drive exploration in RL, and evaluate the quality of these uncertainty estimates in probabilistic image classification and in a regression task predicting synergies of drug combinations.

## 2 Excess Risk, Epistemic Uncertainty, and Model Misspecification

In this section, we analyze the sources of lack of knowledge through the lens of the excess risk of a predictor, define model misspecification, and discuss the pitfall of using discrepancy-based measures of uncertainty when the model is misspecified. The section is divided as follows:

- In Sec. 2.1, we define the mathematical framework and notations used in the analysis.
- In Sec. 2.2, we characterize the sources of lack of knowledge, and argue that estimating the excess risk is a sound measure of epistemic uncertainty. This is the main rationale behind our framework, DEUP, presented in Sec. 3.
- In Sec. 2.3, we briefly analyze the sub-optimal behavior of the Bayesian posterior when the model is misspecified, and explain why in practice models are generally biased, even non-parametric ones.

### 2.1 Notations and Background

Predictive models tackle the problem of learning to predict outputs $y \in \mathcal{Y}$ given inputs $x \in \mathcal{X} \subseteq \mathbb{R}^d$ using a function $f : \mathcal{X} \to \mathcal{A}$ estimated from a training dataset $z^N := (z_1, \ldots, z_N) \in \mathcal{Z}^N$, where $\mathcal{Z} = \mathcal{X} \times \mathcal{Y}$ and $z_i = (x_i, y_i)$, and the unknown ground-truth generative model is defined by $P(X, Y) = P(X)P(Y \mid X)$. In decision theory, the set $\mathcal{A}$ is called the action space, and is typically either equal to $\mathcal{Y}$ for regression problems, or to $\Delta(\mathcal{Y})$, the set of probability distributions on $\mathcal{Y}$, for classification problems. Given a loss function $l : \mathcal{Y} \times \mathcal{A} \to \mathbb{R}^+$ (e.g., $l(y, a) = \|y - a\|^2$ for regression, and $l(y, a) = -\log a(y)$ for classification), the **point-wise risk** (or **expected loss**) of a predictor $f$ at $x \in \mathcal{X}$ is defined as:

$$R(f, x) = \mathbb{E}_{P(Y|X=x)}[l(Y, f(x))]. \tag{1}$$

We define the **risk** (or total expected loss) of a predictor as the marginalization of (1) over $x$:

$$R(f) = \mathbb{E}_{P(X,Y)}[l(Y, f(X))]. \tag{2}$$

Given a hypothesis space $\mathcal{H}$, a subset of $\mathcal{F}(\mathcal{X}, \mathcal{A})$, the set of functions $f : \mathcal{X} \to \mathcal{A}$, the goal of any learning algorithm (or learner) $\mathcal{L}$ is to find a predictive function $h^* \in \mathcal{H}$ with the lowest possible risk:

$$h^* = \underset{h \in \mathcal{H}}{\arg\min} \, R(h). \tag{3}$$

Naturally, the search for $h^*$ is elusive given that $P(X, Y)$ is usually unknown to the learner. Instead, the learner $\mathcal{L}$ maps a finite training dataset $z^N$ to a predictive function $\mathcal{L}(z^N) = h_{z^N} \in \mathcal{H}$ minimizing an approximation of (2), called the **empirical risk**:

$$R_{z^N}(h) = \frac{1}{N} \sum_{i=1}^{N} l(y_i, h(x_i)) \quad \text{where } z_i = (x_i, y_i) \tag{4}$$

$$h_{z^N} = \underset{h \in \mathcal{H}}{\arg\min} \, R_{z^N}(h). \tag{5}$$

The dataset $z^N$ need not be used solely to define the empirical risk as in (4). The learner can for example use a subset of $z^N$ as a validation set to tune its hyperparameters and thus its *effective capacity* (Arpit et al., 2017).

## 2.2 Sources of lack of knowledge

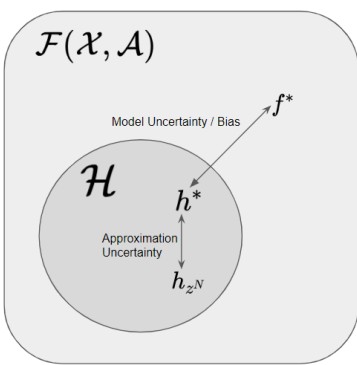

Figure 2: Graphical representations of the two components of epistemic uncertainty. Inspired by Fig. 4 of Hüllermeier & Waegeman (2019).

Clearly, a high value of $R(h_{z^N}, x)$ indicates lack of knowledge around $x$. Similar to Hüllermeier & Waegeman (2019), we characterize the three sources of this lack of knowledge, illustrated in Fig. 2, as follows:

- (1) has a fundamental limiting lower bound, usually reached at a function $f^*$ called the **Bayes predictor**[1]:

$$f^*(x) = \arg\min_{a \in \mathcal{A}} \mathbb{E}_{P(Y|X=x)}[l(Y, a)]. \tag{6}$$

$R(f^*, x) > 0$ indicates an irreducible risk due to the inherent randomness of $P(Y \mid X = x)$. $R(f^*, x)$ is thus a measure of **aleatoric uncertainty** at $x$. Note that there might be more than one Bayes predictor, but by definition, they all have the same point-wise risk, which we denote as $A$:

$$A(x) = R(f^*, x) \tag{7}$$

- Minimizing the risk over $\mathcal{H}$ rather than $\mathcal{F}(\mathcal{X}, \mathcal{A})$ induces a discrepancy between the predictor that is optimal in $\mathcal{H}$, $h^*$, and the Bayes predictor $f^*$, usually referred to as **model uncertainty** (our hypothesis space could be improved). This can be seen as a form of **bias**, as the optimization is limited to functions in $\mathcal{H}$.

- Minimizing the empirical risk instead of the risk induces a discrepancy between $h_{z^N}$ and $h^*$, called the **approximation uncertainty**, due both to the finite training set and finite computational resources for training.

Borrowing terminology from Futami et al. (2022), we use the Bayes predictor to define the excess risk as follows:

**Definition 1** *The **excess risk** of a predictor $f : \mathcal{X} \to \mathcal{A}$ at $x$ is the gap between the point-wise risk and its fundamental limit:*

$$\mathrm{ER}(f, x) = R(f, x) - A(x). \tag{8}$$

Both model uncertainty and approximation uncertainty are indicative of the epistemic state of the learner, and can be reduced with better choices (of $\mathcal{H}$) and more data respectively. This implies that an estimator of the excess risk of $f$ at $x$ (Def. 1) can be used as a measure of **epistemic uncertainty**. ER is intimately linked to the minimum excess risk (Xu & Raginsky, 2020), a measure of the gap between the minimum expected loss attainable by learning from data and $A(x)$, which was shown to decrease and converge towards 0 as the training dataset size grows to infinity, which is a desirable property for EU measures. Using an estimator of ER as a measure of EU is the main idea behind DEUP, and will be expanded upon in Sec. 3.

The following examples illustrate the concepts introduced thus far:

**Example 1** *Consider a univariate regression problem with Gaussian ground truth, i.e. $\mathcal{Y} = \mathbb{R}$ and there exist functions $\mu$ and $\sigma$ such that $P(Y \mid X = x) = \mathcal{N}(Y; \mu(x), \sigma^2(x))$, with the squared loss $l(y, a) = (y - a)^2$. Then, the Bayes predictor is $f^* = \mu$, which has a point-wise risk $x \mapsto A(x) = \sigma^2(x)$, and for any predictor $f : \mathcal{X} \to \mathcal{R}$, and for every $x \in \mathcal{X}$:*

$$R(f, x) = \sigma^2(x) + (f(x) - \mu(x))^2$$
$$\mathrm{ER}(f, x) = (f(x) - \mu(x))^2$$

**Example 2** *Consider a classification problem with $K$ classes, i.e. $\mathcal{Y} = \{1, \ldots, K\}$ and there exists a function $\mu : \mathcal{X} \to \Delta^K$, where $\Delta^K$ is probability simplex of dimension $K - 1$, such that $P(Y \mid X = x)$ is the categorical*

---

[1]An equivalent definition is $f^* = \arg\min_{f \in \mathcal{F}(\mathcal{X}, \mathcal{A})} R(f)$.

*distribution with probability mass function $\mu(x)$, with the log loss $l(y, a) = -\log a(y)$. Then, the Bayes predictor is $f^* = \mu$, which has a point-wise risk $x \mapsto A(x) = \mathbb{H}_{P(Y|X=x)}[Y|x]$, the entropy of the ground-truth label distribution, and for any predictor $f : \mathcal{X} \to \Delta^K$, and for every $x \in \mathcal{X}$:*

$$R(f, x) = \mathbb{H}(\mu(x), f(x))$$
$$\text{ER}(f, x) = D_{\text{KL}}(\mu(x), f(x)),$$

*where $\mathbb{H}(.\ ,\ .)$ and $D_{\text{KL}}(.\ ,\ .)$ denote respectively the cross entropy and the Kullback-Leibler (KL) divergence between two distributions.*

It is clear from the definitions above, and from Fig. 2 that the choice of the subset $\mathcal{H}$ can contribute to higher epistemic uncertainty, as it introduces **misspecification** (Masegosa, 2020; Hong & Martin, 2020; Cervera et al., 2021):

**Definition 2** *A learner with hypothesis space $\mathcal{H} \subseteq \mathcal{F}(\mathcal{X}, \mathcal{A})$ is said to be learning under model misspecification if $f^* \notin \mathcal{H}$, where $f^*$ is the Bayes predictor.*

While there is no agreed upon measure of misspecification, some authors (Masegosa, 2020; Hong & Martin, 2020) focus on Bayesian or approximate Bayesian learners, which maintain a distribution over predictors, and use a discrepancy measure between the best reachable posterior predictive $p(Y \mid X)$ defined by functions $h \in \mathcal{H}$, and the ground-truth likelihood $P(Y \mid X)$. Alternatively, and assuming the function space $\mathcal{F}(\mathcal{X}, \mathcal{A})$ is endowed with a metric, misspecification (or **bias**) can be defined as the distance between $h^*$ and $f^*$.

Additionally, and as argued by many authors (see e.g. Cervera et al. (2021); Knoblauch et al. (2022)), a common implicit assumption in (approximate) Bayesian methods is correct model specification (i.e., there is no bias). In the next subsection, we will briefly review why this assumption is rarely satisfied in practice, and what it entails in terms of uncertainty modelling.

## 2.3 Bayesian uncertainty under model misspecification

Consider a parametric model $p(Y \mid X; \theta)$ and a learner maintaining a distribution over parameters $\theta \in \Theta$, each corresponding to a predictor $h$ in a parametric set of functions $\mathcal{H}$, possibly starting from a prior $p(\theta)$ that would lead to a posterior distribution $p(\theta \mid z^N)$ upon observing data $z^N$. Clearly, the fact that multiple $\theta$'s and corresponding values of $h$ are compatible with the data and the prior indicates lack of knowledge. Because the lack of knowledge indicates where an interactive learner should acquire more information, this justifies the usage of dispersion measures, such as the variance or the entropy of the posterior predictive, as measures of EU.

While such dispersion measures are indicative of approximation uncertainty, they do not account for model misspecification, or bias. The sub-optimal behavior of Bayesian methods with misspecified models has already been studied in terms of the quality of predictions (Grünwald, 2012; Walker, 2013; Grünwald & Van Ommen, 2017) as well as their generalization properties (Masegosa, 2020). These works show that even though the Bayesian posterior predictive concentrates around the best model $p(Y \mid X; \tilde{\theta})$ with minimum KL divergence to the ground truth $P(Y \mid X)$ within $\mathcal{H}$, it does not fit the data distribution perfectly. Additionally, Kleijn & van der Vaart (2012) provide a theoretical analysis of the sub-optimality of the uncertainty estimates under model misspecification. More specifically, the authors derive a misspecified version of the Bernstein-Von Mises theorem, which implies that the Bayesian credible sets are valid confidence sets when the model is well specified, but prove that under misspecification, the credible sets may over-cover or under-cover, and are thus not valid confidence sets. More recently, Cervera et al. (2021) showed that a Bayesian treatment of a misspecified model leads to unreliable epistemic uncertainty estimates in regression settings, while D'Angelo & Henning (2021) showed that uncertainty estimates in misspecified models lead to undesirable behaviors when used to detect OOD examples.

**Sources of model misspecification:** In the Bayesian framework, the hypothesis class $\mathcal{H}$ and the corresponding set of posterior predictive distributions $p(Y \mid X)$ are only implicitly defined, through the choices of the prior $p(\theta)$, the likelihood functions $p(Y \mid X; \theta)$, and the computation budget. Even if *in theory*, the model

is well specified, in the sense that that there exists $\theta^* \in \Theta$ such that $p(Y \mid X; \theta^*) = P(Y \mid X)$[2], *in practice*, the combination of the prior and the finite computational budget limits the set of reachable $\theta$, resulting in a systematic bias (Knoblauch et al., 2022). This behavior is exacerbated in low data regimes, even with modern machine learning models such as neural networks, because the learner may not use all of its capacity due to the implicit (and not fully understood) regularization properties of stochastic gradient descent (SGD, Kale et al. (2021)), explicit regularization, early stopping, or a combination of these. "All models are wrong, but some are useful" - Box (1976).

**The importance of bias in interactive settings:** Naturally, bad uncertainty estimates also have adverse effects on the performances of interactive learners, that use said estimates to guide the acquisition and evaluation of more inputs (Zhou et al., 2003). In fact, it has long been known (Box & Draper, 1959) that model uncertainty (also called bias in Box & Draper (1959)) is just as important as, if not more than, approximation uncertainty (interestingly referred to as variance in Box & Draper (1959)), for the problem of optimal design. Therefore, a learner using approximation uncertainty alone, measured by the variance of the posterior predictive, can be confidently wrong. When the acquisition of more inputs is guided by these predictions, this can slow down the interactive learning process. In Deep Ensembles (Lakshminarayanan et al., 2017) for example, if all the networks in the ensemble tend to fail in a systematic (i.e. potentially reducible) way, this aspect of prediction failure will not be captured by variance, or approximation uncertainty. With flexible models like neural networks however, the hypothesis space $\mathcal{H}$ is defined not only through the chosen architecture of the network, but also through the pre-defined training procedure (e.g. the hyperparameters of the optimizer, the stopping criterion based on the performance on a validation set chosen from the training data...). This means that even though the learning strategy is data-independent, the hypothesis space $\mathcal{H}$ depends on the training data. Arpit et al. (2017) formalized this subtle distinction using the notion of "effective capacity". As a consequence, the learner can incorporate new training examples in areas of the input space where the bias is large, in order to change the hypothesis space $\mathcal{H}$ to $\mathcal{H}'$, which can be closer to the Bayes predictor, thus reducing the bias, and de facto the excess risk ER.

As a consequence of the ubiquity of misspecified models, it is important to question the status quo of relying on discrepancy-based measures of epistemic uncertainty in machine learning, and explore alternative ways of measuring EU, such as DEUP, which we introduce in the next section.

## 3 Direct Epistemic Uncertainty Prediction

As argued in the previous section, the excess risk (8) of a predictor $f$ at a point $x$ can be used as a measure of EU that is robust against model misspecification. Estimating the excess risk $\text{ER}(f, x)$ requires an estimate of the expected loss $R(f, x)$, and an estimate of the aleatoric uncertainty $A(x)$ (7).

DEUP (Direct Epistemic Uncertainty Prediction) **uses observed out-of-sample errors in order to train an error predictor** $x \mapsto e(f, x)$ estimating $x \mapsto R(f, x)$. These may be in-distribution or out-of-distribution errors, depending on what we care about and the kind of data that is available. Given a predictor $x \mapsto a(x)$ of aleatoric uncertainty, $u(f, x)$ defined by:

$$u(f, x) = e(f, x) - a(x), \tag{9}$$

becomes an estimator of $\text{ER}(f, x)$ and as a consequence, an estimator of the epistemic uncertainty of $f$ at $x$. Before delving into the details of DEUP, we briefly discuss how one can obtain an estimator $a$ of aleatoric uncertainty.

**How to estimate aleatoric uncertainty?** We consider three possible scenarios for the aleatoric uncertainty:

1. If we know that $A(x) = 0$, i.e. that the data-generating process is noiseless, then $u(f, x)$ is an estimate of $R(f, x)$, and $a(x)$ can safely be set to 0. This is the case in the experiments we present in Sec. 5.1.

---

[2]Note how we use upper case $P$ for the ground truth posterior and lower case $p$ for the likelihood and the Bayesian posterior predictive estimator.

2. In regression settings with the squared loss, where $A(x)$ equals the variance of $P(Y \mid X = x)$ (Ex. 1), if we have access to an oracle that samples $Y$ given a query $x$ from the environment $P(Y \mid x)$ (e.g., in active learning or SMO), then $A(x)$ can be estimated using the empirical variance of different outcomes of the oracle at the same input $x$. It is common practice for example to perform replicate experiments in biological assays and use variation across replicates to estimate aleatoric uncertainty (Lee et al., 2000; Schurch et al., 2016).

   More formally, if we have multiple independent outcomes $y_1, \ldots, y_K \sim P(Y \mid x)$ for each input point $x$, then training a predictor $a$ with the squared loss on (input, target) examples $\left(x, \frac{K}{K-1}\overline{Var}(y_1, \ldots, y_K)\right)$, where $\overline{Var}$ denotes the empirical variance, yields an estimator of the aleatoric uncertainty.

   Naturally, this estimator is asymptotically unbiased if the learning algorithm ensures asymptotic convergence to a Bayes predictor. This is due to the known fact that

$$\mathbb{E}_{y_1, \ldots, y_K \sim P(Y \mid x)} \left[ \frac{K}{K-1}\overline{Var}(y_1, \ldots, y_K) \right] = Var_{P(Y \mid x)}[Y \mid x]. \tag{10}$$

   We illustrate how such an estimator could be used to obtain EU estimates from the error predictor in Appendix E.

3. In cases where it is not possible to estimate the aleatoric uncertainty, we can use the expected loss estimate $e(f, x)$ as a pessimistic (i.e. conservative) proxy for $u(f, x)$, i.e. set $a(x)$ to 0 in (9). This is particularly relevant for settings when uncertainty estimates are used only to rank different data-points, and in which there is no reason to suspect that there is a significant variability in aleatoric uncertainty across the input space. This is actually the implicit assumption made whenever the variance or entropy of the posterior predictive (which, in principle, accounts for aleatoric uncertainty) is used to measure epistemic uncertainty. This is the case in the experiments we present in Sec. 5.3.

In real life settings, an estimator $x \mapsto a(x)$ of aleatoric uncertainty can be available (e.g. margins of errors of instruments used in a wet lab experiment), and can thus readily be plugged into (9). In the following subsections, we thus assume that $x \mapsto a(x)$ is available, whether it is identically equal to 0 or not, and focus on estimating the point-wise risk or expected loss $R(f, x)$. In Sec. 3.1, we present DEUP in a fixed training set setting, when the learner has access to a held-out validation set. In Sec. 3.2, we demonstrate how the challenges brought about by interactive settings actually bypass the need for the validation set required to train the error predictor.

## 3.1 Fixed Training Set

Using the notations introduced in Sec. 2, we first consider the scenario where we are interested in estimating EU for one predictor $h_{z^N}$, trained on a given training set $z^N$ of a certain size $N$, and a held-out validation set $z'^K$ is available. Recall that the goal is to obtain an estimator $x \mapsto e(h_{z^N}, x)$ of the point-wise risk $x \mapsto R(h_{z^N}, x) = \mathbb{E}_{P(Y \mid x)}[l(Y, h_{z^N}(x))]$.

Each $z'_i = (x'_i, y'_i)$ in the validation dataset can be used to compute an out-of-sample error $e'_i = l(y'_i, h_{z^N}(x'_i))$. Training a secondary predictor $e$ on input-target pairs $(x'_i, e'_i)$, for $i \in \{1, \ldots, K\}$ yields the desired estimator $x \mapsto e(h_{z^N}, x)$. The procedure is summarized in Algorithm 1. It is used for the example provided in Fig. 1

---

**Algorithm 1** DEUP with a fixed training set

**Inputs:** $h = h_{z^N}$, a trained predictor. $a$, an estimator of aleatoric uncertainty. $z'^K = \{(x'_i, y'_i)\}_{i \in \{1, \ldots, K\}}$, a validation set.
**Outputs:** $u : \mathcal{X} \to \mathbb{R}$, an estimator of the EU of $h$.
**Training:**
Create input-error dataset $\mathcal{D}_e = \{(x'_i, l(y'_i, h(x'_i))\}_{i \in \{1, \ldots, K\}}$
Fit an estimator $x \mapsto e(x)$ on $\mathcal{D}_e$, using the squared loss $l(e, e') = (e - e')^2$
**Return:** The predictor $x \mapsto u(x) = e(x) - a(x)$.

---

### 3.2 Interactive Settings

Interactive settings, in which EU estimates are used to guide the acquisition of new examples, provide a more interesting use case for DEUP. However, they bring their own challenges, as the main predictor is retrained multiple times with the newly acquired points. We discuss these challenges below, along with simple ways to mitigate them.

First, as the growing training set $z^N = \{(x_i, y_i)\}_{i \in \{1,\dots,N\}}$ for the main predictor $h_{z^N}$ changes at each acquisition step (as $z^N$ becomes $z^{N+1}$ by incorporating the pair $(x_{N+1}, y_{N+1})$ for example), it is necessary to view the risk estimate $e(h_{z^N}, x)$ as a function of the pair $(x, z^N)$, rather than a function of $x$ only, unlike the fixed training set setting. The error predictor $e$, used to guide acquisition, needs to provide accurate uncertainty estimates as soon as the main predictor changes from $h_{z^N}$ to $h_{z^{N+1}}$ (for clarity, we assume that only one point is acquired at each step, but this is not a requirement for DEUP). This means that $e$ needs to generalize not only over input data $x$, but also over training datasets $z^N$. $e$ should thus be trained using a dataset $\mathcal{D}_e$ of input-target pairs $((x, z^N), l(y, h_{z^N}(x)))$, where $(x, y)$ is not part of $z^N$. This would make the error predictor robust to feedback covariate shift (Fannjiang et al., 2022), i.e. the distribution shift resulting from the acquired points being a function of the training data. Conditioning on the dataset explicitly also address concerns from Bengs et al. (2022) which shows that directly estimating uncertainty using a second order predictor trained simply with input data using L2 error yields poor uncertainty estimates. A learning algorithm with good generalization properties (such as modern neural networks), would in principle be able to extrapolate from $\mathcal{D}_e$, and estimate the errors made by a predictor $h$ on points $x \in \mathcal{X}$ not seen so far, i.e. belonging to what we call the *frontier of knowledge*.

Second, the learner usually does not have the luxury of having access to a held-out validation set, given that the goal of such an interactive learner is to learn the Bayes predictor $f^*$ using as little data as possible. This makes Algorithm 1 inapplicable to this setting. While it might be tempting to replace the held-out set $z'^K$ in Algorithm 1 with the training set $z^N$, this would lead to an error predictor trained with in-sample errors rather that out-of-sample errors, thus severely limiting its ability to generalize its resulting uncertainty estimates out-of-sample and generally underestimating EU.

Denoting by $N_0 \geq 0$ the number of initially available training points before any acquisition, and observing that for $i > N_0$, the pair $(x_i, y_i)$ is not used to train the predictors $h_{z^{N_0}}, h_{z^{N_0+1}}, \dots, h_{z^{i-1}}$, we propose to use the future acquired points as out-of-sample examples for the past predictors, in order to build the training dataset $\mathcal{D}_e$ for the error estimator. At step $M > N_0$, i.e. after acquiring $(M - N_0)$ additional input-output pairs $(x, y)$ and obtaining the predictor $h_{z^M}$, $\mathcal{D}_e$ is equal to:

$$\mathcal{D}_e = \bigcup_{i=N_0+1}^{M} \bigcup_{N=N_0}^{i-1} \{((x_i, z^N), l(y_i, h_{z^N}(x_i)))\}. \tag{11}$$

Using $\mathcal{D}_e$ (11) requires storing in memory all versions of the main predictor $h$ (i.e. $h_{z^{N_0}}, \dots, h_{z^M}$), which is impractical. Additionally, it requires using predictors that take as an input a dataset of arbitrary size, which might lead to overfitting issues as the dataset size grows. Instead, we propose the following two approximations of $\mathcal{D}_e$:

1. We embed each input pair $(x_i, z^N)$ in a feature space $\Phi$, and replace each such pair in $\mathcal{D}_e$ with the feature vector $\phi_{z^N}(x)$, hereafter referred to as the **stationarizing features** of the dataset $z^N$ at $x$.

2. To alleviate the need of storing multiple versions of $h$, we make each pair $(x_i, y_i)$ contribute to $\mathcal{D}_e$ once rather than $i - N_0$ times, by replacing the inner union of (11) with the singleton $\{((x_i, z^{i-1}), l(y_i, h_{z^{i-1}}(x_i)))\}$. Said differently, for each predictor $h_{z^N}$, only the next acquired point $(x_{N+1}, y_{N+1})$ is used to populate $\mathcal{D}_e$.

These approximations result in the following training dataset of the error estimator at step $M$:

$$\mathcal{D}_e = \{(\phi_{z^{i-1}}(x_i), l(y_i, h_{z^{i-1}}(x_i)))\}_{i \in \{N_0+1,\dots,M\}}. \tag{12}$$

---

**Algorithm 2** Pre-filling the uncertainty estimator training dataset $\mathcal{D}_e$

---

$\mathcal{D}_e \leftarrow \emptyset$
**while** $|\mathcal{D}_e| < N_{pretrain}$ **do**
  Split $\mathcal{D}_{init} = z^{N_0}$ into $K$ random subsets $\mathcal{D}_1, \ldots, \mathcal{D}_K$ of equal size. Define $\tilde{\mathcal{D}} = \bigcup_{k=1}^{K-1} \mathcal{D}_k$.
  Fit a new predictor $h_{\tilde{\mathcal{D}}}$ on $\tilde{\mathcal{D}}$, and fit the features $\phi_{\tilde{\mathcal{D}}}$ on $\tilde{\mathcal{D}}$.
  $\mathcal{D}_e \leftarrow \mathcal{D}_e \cup \bigcup_{(x,y) \in \mathcal{D}_K} \{(\phi_{\tilde{\mathcal{D}}}(x), l(y, h_{\tilde{\mathcal{D}}}(x)))\}$.
**end**

---

In this paper, we explored $\phi_{z^N}(x) = \left(x, s, \hat{q}(x \mid z^N), \hat{V}(\tilde{\mathcal{L}}, z^N, x)\right)$ or a subset of these 4 features, where $\hat{q}(x \mid z^N)$ is a density estimate from data $z^N$ at $x$, $s = 1$ if $x$ is part of $z^N$ and otherwise 0, $\tilde{\mathcal{L}}$ a learner that produces a distribution over predictors, e.g. a GP or an ensemble of neural networks (Lakshminarayanan et al., 2017)), and $\hat{V}(\tilde{\mathcal{L}}, z^N, x)$ is an estimate of the model variance of $\tilde{\mathcal{L}}(z^N)$ at $x$. Note that $\tilde{\mathcal{L}}$ can be chosen to be the same as $\mathcal{L}$ (Sec. 2). For numerical reasons, we found it preferable to use $\log \hat{q}$ and $\log \hat{V}$ instead of $\hat{q}$ or $\hat{V}$ as input features. $\hat{q}$ can be obtained by training a density estimator (such as a Kernel Density Estimator or a flow-based deep network (Rezende & Mohamed, 2015)). Like the other predictors, the density estimator also needs to be fine-tuned when new data is added to the training set. While these features are not required per se to train DEUP, they provide clues to help training the uncertainty estimator, and one can play with the trade-off of computational cost versus usefulness of each clue. They also provide DEUP the flexibility to incorporate useful properties. For instance, the density can help in incorporating *distance awareness* which Liu et al. (2020) show is critical for uncertainty estimation. They sometimes come at no greater cost, if our main predictor is the mean prediction of the learner's output distribution, and if we use the corresponding variance as the only extra feature, as is the case in the experiments of Sec. 5.1 with GPs. As (12) is merely an approximation of (11), not all subsets of features are expected to be useful in all settings. In the ablations presented in Appendices A and C, we experimentally confirm that $x$ alone is not sufficient, and the stationarizing features are critical for reliable uncertainty estimates. Additionally, we observe that having $x$ as part of $\phi_{z^N}$ can help in some tasks and hurt in others. We note that the choice of features here is a design choice, and the usage of other features could be investigated in future work.

**Pre-training the error predictor:** If the learner cannot afford to wait for a few rounds of acquisition in order to build a dataset $\mathcal{D}_e$ large enough to train the error predictor (e.g. when the prediction target $y$ is the output of a costly oracle), it is possible to pre-fill $\mathcal{D}_e$ using the $N_0$ initially available training points $z^{N_0}$ only, following a strategy inspired by $K-$fold cross validation. We present one such strategy in Algorithm 2. The procedure stops when the training dataset for the secondary learner, $\mathcal{D}_u$, contains at least $N_{pretrain}$ elements. In our experiments, we choose $N_{pretrain}$ to be a small multiple of the number of initial training points.

Putting all these things together yields the pseudo-code for DEUP in interactive learning settings provided in Algorithm 3. In practice, in addition to out-of-sample errors, the secondary predictor can be trained with in-sample errors, which inform the error predictor that the uncertainty at the training points should be low. The stopping criterion is defined by the interactive learning task. It can be a fixed number of iterations for example, a criterion defined by a metric evaluated on a validation set, or a criterion defined by the optimal value reached (maximum or mimumum) in SMO. The algorithm is agnostic to the acquisition function or policy $\pi(. \mid h, u)$, the acquisition machinery that proposes new input points from $\mathcal{X}$, using the current predictor $h$ and its corresponding EU estimator $u$. Examples of acquisition functions in the context of SMO include Expected Improvement (Močkus, 1975) and Upper Confidence Bound (UCB, Srinivas et al. (2010)).

## 4 Related Work

**Bayesian Learning.** Bayesian approaches have received significant attention as they provide a natural way of representing epistemic uncertainty in the form of the Bayesian posterior distributions. Gaussian Processes (Williams & Rasmussen, 1995) are a popular way to estimate EU, as the variance among the functions in the posterior (given the training data) can be computed analytically. Beyond GPs, efficient MCMC-based approaches have been used for approximating samples from the Bayesian posterior on large

---

**Algorithm 3** Training procedure for DEUP in an Interactive Learning setting

---

**Inputs:**    $\mathcal{D}_{init} = z^{N_0} = \{(x_i, y_i)\}_{i \in \{1, \ldots, N_0\}}$. $a$, an estimator of aleatoric uncertainty. $\Phi$, the embedding space (i.e. the chosen stationarizing features). $\pi$, the acquisition machinery.
$\mathcal{D}_e \leftarrow \emptyset$, training dataset for the error predictor $e$
$\mathcal{D} \leftarrow \mathcal{D}_{init}$, dataset of training points for the main predictor
$x_{acq} \leftarrow \emptyset, \ \ y_{acq} \leftarrow \emptyset$
**Optional:** Pre-fill $\mathcal{D}_e$ using Algorithm 2
**while** *stopping criterion not reached* **do**

    Fit predictor $h_{\mathcal{D}}$ and features $\phi_{\mathcal{D}}$ on $\mathcal{D}$
    Fit a predictor $\phi \mapsto e(\phi)$ on $\mathcal{D}_e$
    $x_{acq} \sim \pi(. \mid h_{\mathcal{D}}, x \mapsto e(\phi_{\mathcal{D}}(x)) - a(x))$ (can be either a single point, or a batch of points)
    Sample outcomes from the ground truth distribution: $y_{acq} \sim P(. \mid x_{acq})$
    $\mathcal{D}_e \leftarrow \mathcal{D}_e \cup \{(\phi_{\mathcal{D}}(x_{acq}), l(y_{acq}, h_{\mathcal{D}}(x_{acq})))\}$
    $\mathcal{D} \leftarrow \mathcal{D} \cup \{(x_{acq}, y_{acq})\}$
**end**

---

datasets (Welling & Teh, 2011; Zhang et al., 2020; Vadera et al., 2020a). In the deep learning context, Blundell et al. (2015); Kendall & Gal (2017); Depeweg et al. (2018) use the posterior distribution of weights in Bayesian Neural Networks (BNNs) (MacKay, 1992) to capture EU. SWAG (Maddox et al., 2019) fits a Gaussian distribution on the first moments of SGD iterates building upon SWA (Izmailov et al., 2018) to define the posterior over the neural network weights. This distribution is then used as a posterior over the neural network weights. Dusenberry et al. (2020) parametrize the BNN with a distribution on a rank-1 subspace for each weight matrix. Other techniques that rely on measuring the discrepancy between different predictors as a proxy for EU include MC Dropout (Gal & Ghahramani, 2016), that interprets Dropout (Hinton et al., 2012) as a variational inference technique in BNNs. These approaches, relying on sampling of weights or dropout masks at inference time, share some similarities with ensemble-based methods, that include bagging (Breiman, 1996) and boosting (Efron & Tibshirani, 1994), where multiple predictors are trained and their outputs are averaged to make a prediction, although the latter measure variability due to the training set instead of the spread of functions compatible with the given training set, as in Bayesian approaches. Deep Ensembles (Lakshminarayanan et al., 2017) are closer to the Bayesian approach, using an ensemble of neural networks that differ because of randomness in initialization and training (as you would have with MCMC, albeit in a more heuristic way). Wen et al. (2020) present a memory-efficient way of implementing deep ensembles, by using one shared matrix and a rank-1 matrix for the parameters per member, while Vadera et al. (2020b); Malinin et al. (2020) improve the efficiency of ensembles by distilling the distribution of predictions rather than the average, thus preserving the information about the uncertainty captured by the ensemble. Classical work on *Query by committee* (Seung et al., 1992; Freund et al., 1992; Gilad-Bachrach et al., 2005; Burbidge et al., 2007) also studied the idea of using discrepancy as a measure for information gain for the design of experiments. Tagasovska & Lopez-Paz (2019) propose using orthonormal certificates which capture the distance between a test sample and the dataset. Liu et al. (2020) further establish the importance of this notion of *distance awareness* for uncertainty estimation, and along with proceeding methods DUE (van Amersfoort et al., 2021), and DDU (Mukhoti et al., 2021) combine feature representations learnt by deep neural networks with exact Bayesian inference methods like GPs and Gaussian Discriminant Analysis. This line of work falls in the umbrella of Deep Kernel Learning (Wilson et al., 2016, DKL;). van Amersfoort et al. (2020) present an alternative instantiation of DKL using RBF networks (LeCun et al., 1998). DUN (Antoran et al., 2020) uses the disagreement between the outputs from intermediate layers as a measure of uncertainty. Another line of work that has received attention in the community recently is that of evidential uncertainty estimation (Malinin & Gales, 2018; Sensoy et al., 2018; Amini et al., 2020) which estimates EU based on a parameteric estimate of the model variance, which has been shown to have poor uncertainty estimates in recent work by Bengs et al. (2022). Finally, Tran et al. (2022) combine several of these techniques in the context of large neural networks. Also closely related to the stationarizing features introduced in Sec. 3.2, Morningstar et al. (2021); Haroush et al. (2021) use estimators of statistical features to capture the uncertainty for OOD detection.

**Distribution-Free Uncertainty Estimation.** Conformal Prediction (Vovk et al., 2005; Shafer & Vovk, 2008; Angelopoulos & Bates, 2021) is an alternative to Bayesian methods for uncertainty estimation. Conformal prediction involves building statistically rigorous uncertainty sets and intervals for model predictions, which are *guaranteed* to contain the ground truth with a specified probability. It is also closely linked to the statistical paradigm of hypothesis testing (Angelopoulos & Bates, 2021). Conformal prediction is an appealing alternative to Bayesian approaches as it can be applied on top of existing models, and does not require any special training procedure. Recent work has demonstrated the efficacy of applying conformal prediction with neural network models for time series (Lin et al., 2022; Zaffran et al., 2022) and image data (Angelopoulos et al., 2020). Fannjiang et al. (2022) study conformal prediction within an active learning setting. While DEUP can broadly be categorized as a distribution-free uncertainty estimation methods, it differs from Conformal Prediction as it does not require a pre-defined degree of confidence before outputting a prediction set.

**Loss Prediction.** Kull & Flach (2015) present several decompositions of the total expected loss, including the decomposition into the epistemic and irreducible (aleatoric) loss. They present additive adjustments that reduce the scoring rules like the log-loss and Brier score, but do not tackle the general problem of uncertainty estimation. More closely related to DEUP, Yoo & Kweon (2019) propose a loss prediction module for learning to predict the value of the loss function. Hu et al. (2020) also propose using a separate network that learns to predict the variance of an ensemble. These methods, however, are trained only to capture the in-sample error, and do not capture the out-of-sample error which is more relevant for scenarios like active learning where we want to pick $x$ where the reducible *generalization error* is large. EpiOut (Umlauft et al., 2020; Hafner et al., 2019) propose learning a binary output that simply distinguishes between low or high EU.

## 5 Experiments

Through the experiments described below, we aim to provide evidence for the following key claims: (**C1**) Epistemic uncertainty measured by DEUP leads to significant improvements in downstream decision making tasks compared to established baselines, and (**C2**) The error predictor learned by DEUP can generalize to unseen samples. We emphasize that in order to make fair comparisons, **DEUP does not have access to any additional OOD data during training**, in all experiments presented in this section. Instead, when required, we use Algorithm 2 to generate the OOD data used for training the error predictor. Finally, note that in terms of computational cost, training DEUP with density and model variance as stationarizing features is on-par with training an ensemble of 5 networks.

### 5.1 Sequential Model Optimization

Sequential model optimization is a form of interactive learning, where at each stage, the learner chooses query examples to label, looking for examples with a high value of the unknown oracle function. Such examples are selected so they have a high predicted value (to maximize the unknown oracle function) and a large predicted uncertainty (offering the opportunity of discovering yet higher values). Acquisition functions, such as Expected Improvement (EI, Močkus (1975)) trade-off exploration and exploitation, and one can select the next candidate by looking for $x$'s maximizing the acquisition function. We combine EI with DEUP (and call the method DEUP-EI, by analogy to GP-EI, a common SMO baseline), treating the main predictor and DEUP EU predictions at $x$ respectively as mean and standard deviation of a Gaussian distribution for the learner's guess of the value of the oracle at $x$. Similarly, when using MC-Dropout or Deep Ensembles as predictors, we refer to the resulting methods as MCDropout-EI and Ensembles-EI respectively.

We first consider in Fig. 3 (Left) a synthetic one-dimensional function with multiple local maxima, and starting from an initial dataset of 6 pairs $(x, y)$, we compare the maximum values reached by each of DEUP-EI, GP-EI, MCDropout-EI, and Ensembles-EI to a random acquisition strategy (which is an unsurprisingly strong baseline given the number of local optima of the target function). Because MCDropout and Ensembles are trained on in-sample data only, they are unable to generalize their uncertainty estimates, which makes them bad candidates for Sequential Model Optimization, because they are easily stuck in local minima, and require many iterations before the acquisition function gives more weight to the predicted uncertainties than the current maximum. For DEUP-EI, the main predictor is a neural network, and the error predictor is a GP

regressor. The stationarizing features $\phi_z(x)$ used are the input $x$ itself and the variance of a GP fit on the available data at every step. More details are provided in Appendix A.1. In Appendix A.2, we show a similar experiment on a two-dimensional function. In Appendix A.4, we provide an ablation study of different subsets of the features $\phi_z(x)$, confirming the necessity of extra features and the usefulness of the input $x$ as well.

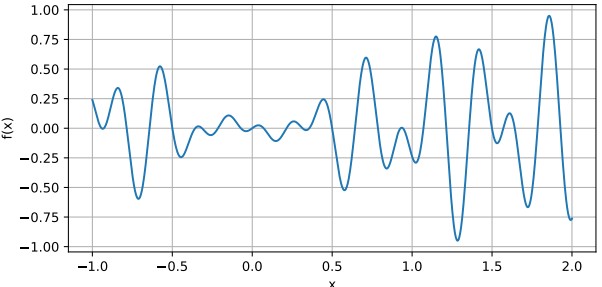 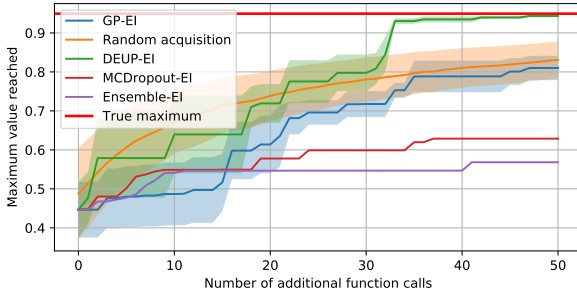

Figure 3: *Left.* Synthetic function to optimize. *Right.* Maximum value reached by the different methods on the synthetic function. The shaded areas represent the standard error across 5 different runs, with different initial sets of 6 pairs. For clarity, the shaded areas are omitted for the two worst performing methods. In each run, all the methods start with the same initial set of 6 points. GP-EI tends to get stuck in local optima and requires more than 50 steps, on average, to reach the global maximum.

Next, we showcase how using DEUP to calibrate GP variances (used as the only input for the error predictor) allows for better performances in higher-dimensional optimization tasks. Specifically, we compare DEUP-EI to TuRBO-EI (Eriksson et al., 2019), a state-of-the-art method for sequential optimization, that fits a collection of local GP models instead of a global one in order to perform efficient high-dimensional optimization, on the Ackley function (Ackley, 2012), a common benchmark for optimization algorithms. This function can be defined for arbitrary dimensions, and has many local minima.

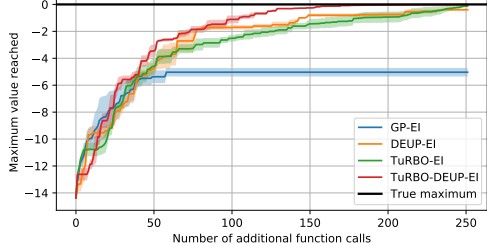 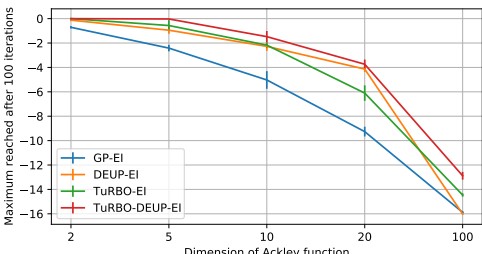

Figure 4: *Left.* Max. value reached by the different optimization methods, for the 10 dimensional Ackley function. In each run, all the methods start with the same initial 20 points. Shaded areas represent the standard error across 3 runs. *Right.* Max. value reached in the budget-constrained setting, on the Ackley functions of different dimensions. Error bars represent the standard error across 3 different runs, with different initial sets of 20 pairs. The budget is 120 function calls in total. Higher is better and TuRBO-DEUP-EI is less hurt by dimensionality.

In Fig. 4 (Left), we compare the different methods on the 10-D Ackley function, and observe that while GP-EI gets stuck in local optima, DEUP-EI is able to reach the global maximum consistently. In Fig. 4 (Right), we show that for budget-constrained optimization problems, adapting DEUP to TuRBO (called TuRBO-DEUP-EI) consistently outperforms regular TuRBO-EI, especially in higher dimensions. More details about the Ackley function and this experiment are provided in Appendix A.3. See also Appendix A for 1D and 2D SMO tasks where DEUP-EI outperforms GP-EI (Bull, 2011), as well as neural networks with MC-Dropout or Ensembles. The experiments presented in this section thus validate our experimental claim **C1**.

## 5.2 Reinforcement Learning

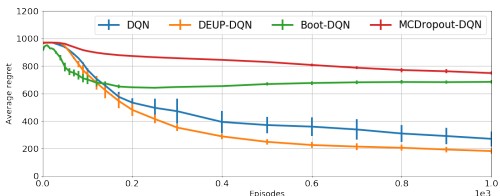

Figure 5: Average regret on CartPole task. Error bars represent standard error across 5 runs.

Similar to SMO, a key challenge in RL is efficient exploration of the input state space. To investigate the effectiveness of DEUP's uncertainty estimates in the context of RL, we present a proof-of-concept for incorporating epistemic uncertainties predicted by DEUP to DQN (Mnih et al., 2013), which we refer to as DEUP-DQN. Specifically, we train the uncertainty predictor with the objective to predict the TD-error, using log-density estimates as a stationarizing feature. The predicted uncertainties are then used as an exploration bonus in the Q-values. As explained in Sec. 3.2, it is the acquired points, before they are used to retrain the main predictor, that act as the *out-of-sample* examples to train DEUP. In RL, because the targets (e.g. of Q-Learning) are themselves estimates and moving, data seen at any particular point is normally out-of-sample and can inform the uncertainty estimator, when the inputs are used with the stationarizing features. Details of the experimental setup are in Appendix B. We evaluate DEUP-DQN on CartPole, a classic RL task from *bsuite* (Osband et al., 2020), against DQN + $\epsilon$-greedy, DQN + MC-Dropout (Gal & Ghahramani, 2016) and Bootstrapped DQN (Osband et al., 2016). Fig. 5 shows that DEUP achieves lower regret faster, compared to all the baselines, which demonstrates the advantage of DEUP's uncertainty estimates for efficient exploration, confirming our claim **C1**. Future work should investigate ways to scale this method to more complex environments.

### 5.3 Uncertainty Estimation

#### 5.3.1 Epistemic Uncertainty Estimation for Drug Combinations

We validate DEUP's ability to generalize its uncertainty estimate (claim **C2**) in a real-world regression task predicting the synergy of drug combinations. While much effort in drug discovery is spent on finding novel small molecules, a potentially cheaper method is identifying combinations of pre-existing drugs which are synergistic (i.e., work well together). However, every possible combination cannot be tested due to the high monetary cost and time required to run experiments. Therefore, developing good estimates of EU can help practitioners select experiments that are both informative and promising. we used the DrugComb and LINCS L1000 datasets (Zagidullin et al., 2019; Subramanian et al., 2017). DrugComb is a dataset consisting of pairwise combinations of anti-cancer compounds tested on various cancer cell lines. For each combination, the dataset provides access to several synergy scores, each indicating whether the two drugs have a synergistic or antagonistic effect on cancerous cell death. LINCS L1000 contains differential gene expression profiles for various cell lines and drugs. Differential gene expressions measure the difference in the amount of mRNA related to a set of influential genes before and after the application of a drug. Because of this, gene expressions are a powerful indicator of the effect of a single drug at the cellular level. As shown in Fig. 6(c), the out-of-sample error predicted by DEUP correlates better with residuals of the model on the test set in comparison to several other uncertainty estimation methods. Moreover, DEUP better captured the order of magnitude of the residuals as shown in Fig. 6, confirming the claim **C2**. Details on experiments and metrics are in Appendix D.

#### 5.3.2 Epistemic Uncertainty Predictions for Rejecting Difficult Examples

Epistemic uncertainty estimates can be used to reject difficult examples where the predictor might fail, such as OOD inputs[3]. We thus consider a standard OOD Detection task (Liu et al., 2020; van Amersfoort et al., 2021; Nado et al., 2021), where we train a ResNet (He et al., 2016) model for CIFAR-10 classification (Krizhevsky, 2009) and reject OOD examples using the estimated uncertainty in the prediction. To facilitate rejection of classes other than those in the training set, we use a Bernoulli Cross-Entropy Loss for each class following van Amersfoort et al. (2020): $l(\hat{f}(x), y) = -\sum_i y_i \log \hat{f}_i(x) + (1 - y_i) \log(1 - \hat{f}_i(x))$, where $y$ is a one-hot vector ($y_i = 1$ if $i$ is the correct class, and 0 otherwise), and $\hat{f}_i(x)$ = predicted probability for class $i$. The target for out-of-distribution data (from *other* classes) can be represented as $y = \{0, \ldots, 0\}$. We use Algorithm 1 with stationarizing features from Sec. 3.2 for training DEUP on this task. At inference time, we can reject an

---

[3]e.g. rare but challenging inputs can be directed to a human, avoiding a costly mistake

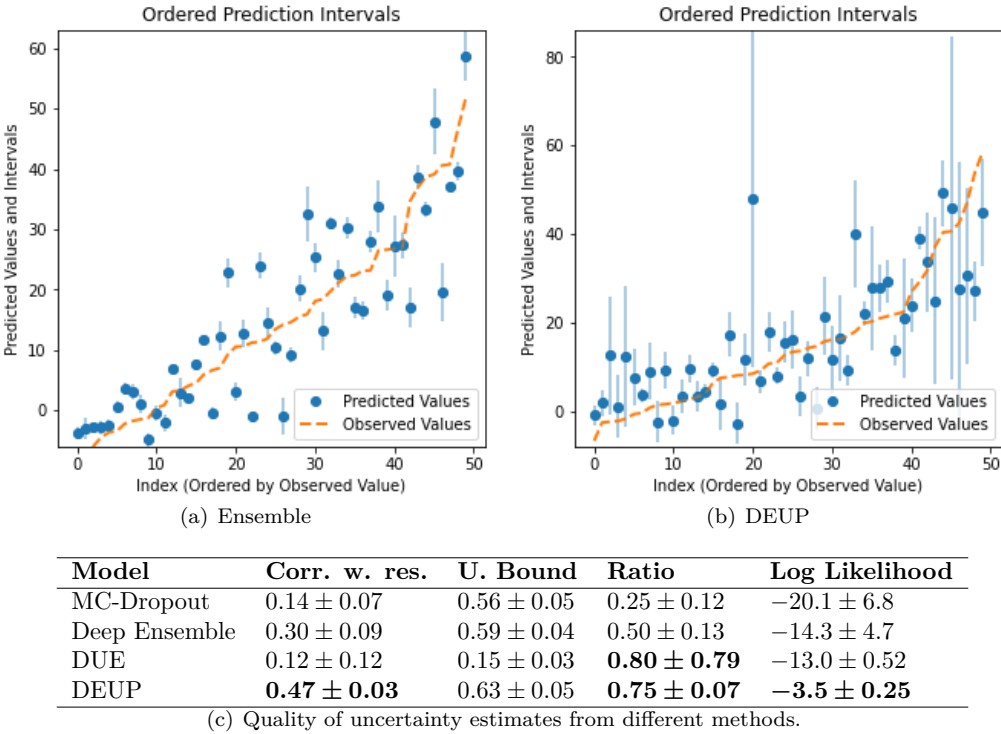

(a) Ensemble

(b) DEUP

| Model | Corr. w. res. | U. Bound | Ratio | Log Likelihood |
|---|---|---|---|---|
| MC-Dropout | $0.14 \pm 0.07$ | $0.56 \pm 0.05$ | $0.25 \pm 0.12$ | $-20.1 \pm 6.8$ |
| Deep Ensemble | $0.30 \pm 0.09$ | $0.59 \pm 0.04$ | $0.50 \pm 0.13$ | $-14.3 \pm 4.7$ |
| DUE | $0.12 \pm 0.12$ | $0.15 \pm 0.03$ | $\mathbf{0.80 \pm 0.79}$ | $-13.0 \pm 0.52$ |
| DEUP | $\mathbf{0.47 \pm 0.03}$ | $0.63 \pm 0.05$ | $\mathbf{0.75 \pm 0.07}$ | $\mathbf{-3.5 \pm 0.25}$ |

(c) Quality of uncertainty estimates from different methods.

Figure 6: Drug Combinations. Predicted mean and uncertainty (error bars) on 50 test examples ordered by increasing value of true synergy score (orange). Model predictions and uncertainties in blue. Ensemble **(a)** (and MC-dropout, not shown) consistently underestimate uncertainty while DEUP **(b)** captures the right order of magnitude. **(c)** *Corr. w. res.* shows correlation between model residuals and predicted uncertainties $\hat{\sigma}$. A best-case *Upper Bound* on *Corr. w. res.* is obtained from the correlation between $\hat{\sigma}$ and true samples from $\mathcal{N}(0, \hat{\sigma})$. *Ratio* is the ratio between col. 1 and 2 (larger is better). *Log-likelihood*: average over 3 seeds of per sample predictive log-likelihood.

example based on the estimated epistemic uncertainty. To ascertain how well an epistemic error estimate sorts unseen examples by the above NLL loss, we consider the rank correlation between the predicted uncertainty and the observed generalization error. Note that we can compute the generalization error for OOD examples with $y = \{0, \ldots, 0\}$ directly since we are using a Binary Cross-Entropy loss. We use examples from SVHN (Netzer et al., 2011) as the OOD examples. This metric focuses on the quality of the uncertainty estimates rather than just their ability to simply classify in- vs out-of-distribution examples. This metric is also an indicator of how accurate the uncertainty estimates are for out-of-distribution examples. We also report the standard AUROC for the OOD detection task. We use MC-Dropout (Gal & Ghahramani, 2016), Deep Ensembles (Lakshminarayanan et al., 2017), DUE (van Amersfoort et al., 2021) and DUQ (van Amersfoort et al., 2020) as the baselines. As the primary focus of our work is estimating epistemic uncertainty, we do not consider specialized methods for OOD detection (Morningstar et al., 2021; Haroush et al., 2021). Additionally, to study the effect of model capacity we consider ResNet-18 and ResNet-50 as the main predictors for all the methods. Additional training details along with results for ResNet-50 are presented in Appendix C.

Table 1 shows, supporting experimental claim **C2**, that with the variance from DUE (van Amersfoort et al., 2021) and the density from MAF (Papamakarios et al., 2017) as stationarizing features, DEUP provides uncertainty estimates that have high rank correlation with the underlying generalization error on OOD data. We also achieve competitive AUROC with the strong baselines, demonstrating that uncertainty estimates from DEUP result in better performance on the downstream task of OOD detection. Additionally, since the error predictor is trained separately from the main predictor, there is no explicit trade-off between the accuracy of the main predictor and the quality of uncertainty estimates. We achieve competitive accuracy of 93.89% for the main predictor. We ignore the effect of aleatoric uncertainty (due to inconsistent human

| Model | SRCC | AUROC |
|---|---|---|
| MC-Dropout | $0.287 \pm 0.002$ | $0.894 \pm 0.008$ |
| Deep Ensemble | $0.381 \pm 0.004$ | $\mathbf{0.933 \pm 0.008}$ |
| DUQ | $0.376 \pm 0.003$ | $0.927 \pm 0.013$ |
| DUE | $0.378 \pm 0.004$ | $0.929 \pm 0.005$ |
| DEUP (D+V) | $\mathbf{0.426 \pm 0.009}$ | $\mathbf{0.933 \pm 0.010}$ |

Table 1: Spearman Rank Correlation Coefficient (SRCC) between predicted uncertainty and OOD generalization error (SVHN); Area under ROC Curve (AUROC) for OOD Detection (SVHN) with CIFAR-10 ResNet-18 models (3 seeds). DEUP significantly outperforms baselines in terms of SRCC and is equivalent to Deep Ensembles but scoring better than the other methods in terms of the coarser AUROC metric.

labelling), which would require a human study to ascertain. We note that we choose the DUE baseline as it is representative of related methods such as SNGP (Liu et al., 2020) and DDU (Mukhoti et al., 2021), and performs best in our experiments. We present additional results in a distribution shift setting in Appendix C. Note that in the pretraining phase of the uncertainty estimator (Algorithm 2), we obtain the subsets by splitting the data based on classes, with each split containing $\lfloor n/K \rfloor$ classes. So when we train on $K-1$ subsets, the $\lfloor n/K \rfloor$ classes from the remaining subset become *out-of-distribution*.

## 6  Conclusion and Future Work

Whereas standard measures of epistemic uncertainty focus on variance (due to approximation error), we argue that bias (introduced by misspecification) should also be accounted for as part of the epistemic uncertainty, as it is reducible for predictors like neural networks whose effective capacity is a function of the training data. By analyzing the sources of lack of knowledge, we describe how the excess risk is a sounds measure of epistemic uncertainty. This motivates the DEUP framework, where we train a second model to predict the risk of the main predictor. In interactive settings, this nonetheless raises non-stationarity challenges for this estimator, and we propose extra input features to tackle this issue, and show experimentally their advantages. Future work should investigate ways to improve the computational efficiency of DEUP (for instance by considering different choices of stationarizing features or alternate ways of incorporating the dataset as input to the secondary predictor), and ways to estimate aleatoric uncertainty when no estimator thereof is readily available and when one cannot simply query an oracle several times on the same input $x$. Theoretically studying the properties of DEUP under model misspecification is another important direction to explore in future work.

## Acknowledgements

The authors would like to thank Tristan Deleu, Anirudh Goyal, Tom Bosc, Chritos Tsirigotis, Léna Néhale Ezzine, Doina Precup, Pierre Luc-Bacon, John Bradshaw, José Miguel Hernández-Lobato as well as the anonymous reviewers for useful comments and feedback. This research was enabled in part by support provided by Compute Canada, the Bill & Melinda Gates Foundation, IVADO and a CIFAR AI Chair. The authors also acknowledge funding from Samsung, IBM, Microsoft.

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

## A Sequential Model Optimization Experiments

For all our Sequential Optimization algorithms, we use Algorithm 3 to train DEUP uncertainty estimators. We found that the optional step of pre-filling the uncertainty estimator dataset $\mathcal{D}_e$ was important given the low number of available training points. We used half the initial training set (randomly chosen) as in-sample examples (used to train the main predictor and an extra-feature generator) and the other half as out-of-sample examples to provide instances of high epistemic uncertainty to train an uncertainty predictor; we repeated the procedure by alternating the roles of the two halves of the dataset. We repeated the whole procedure twice using a new random split of the dataset, thus ending up with 4 training points in $\mathcal{D}_e$ for every initial training point in $\mathcal{D}_{init}$.

The error predictor is trained with the log targets (i.e. log MSE between predicted and observed error). This helps since the scale of the errors varies over multiple orders of magnitude.

Computationally, the training time of DEUP-EI depends on various choices (e.g. the features used to train the epistemic uncertainty predictor, the dimension of the input, the learning algorithms, etc..). Additionally, the training time for the uncertainty predictor varies at each step of the optimization. In total, the sequential optimization experiments took about 1 CPU day.

We use BoTorch[4] (Balandat et al., 2020) as the base framework for our experiments.

### A.1 One-dimensional function

For Random acquisition, we sampled for different seeds 56 points, and used the (average across the seeds of the) maximum of the first 6 values as the first value in the plots (Figs. 3 and 4). Note that because the function is specifically designed to have multiple local maxima, GP-EI also required more optimization steps, and actually performed worse than random acquistion.

As a stationarizing input feature, we used the variance of a GP fit on the available data at every step. We found that the binary (in-sample/out-of-sample) feature and density estimates were redundant with the variance feature and didn't improve the performance as captured by the number of additional function calls. We used a GP for the DEUP uncertainty estimator. Using a neural net provided similar results, but was computationally more expensive in this 1-D case with few datapoints. We used a 3-hidden layer neural network, with 128 neurons per layer and a ReLU activation function, with Adam (Kingma & Ba, 2015) and a learning rate of $10^{-3}$ (and default values for the other hyperparameters) to train the main predictor for DEUP-EI (in order to fit the available data). The same network architecture and learning rate were used for the Dropout and Ensemble baselines. We used 3 networks for the Ensemble baseline, and a dropout probability of 0.3 for the Dropout baseline, with 100 test-time forward passes to compute uncertainty estimates.

---

[4]https://botorch.org/

## A.2 Two-dimensional function

To showcase DEUP's usefulness for Sequential Model Optimization in with a number of dimensions greater than 1, we consider the optimization of the Levi N.13 function, a known benchmark for optimization. The function $f$ takes a point $(x, y)$ in 2D space and returns:

$$f(x, y) = - \left( \sin^2(3\pi x) + (x - 1)^2(1 + \sin^2(3\pi y)) + (y - 1)^2(1 + \sin^2(2\pi y)) \right)$$

We use the box $[-10, 10]^2$ as the optimization domain. In this domain, the maximum of the function is 0, and it is reached at $(1, 1)$. The function has multiple local maxima, as shown in Fig. 7(a)[5].

Similar to the previous one-dimensional function, MCDropout and Ensemble provided bad performances and are omitted from the plot in Fig. 7(b). We used the same setting and hyperparameters for DEUP as for the previous function. DEUP-EI is again the only method that reaches the global maximum consistently in under 56 function evaluations.

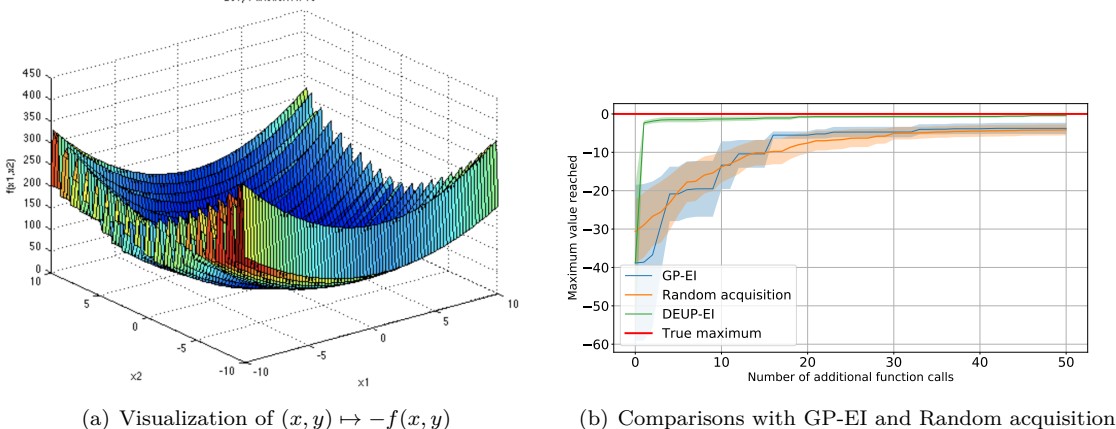

(a) Visualization of $(x, y) \mapsto -f(x, y)$      (b) Comparisons with GP-EI and Random acquisition

Figure 7: Sequential Model Optimization on the Levi N.13 function

## A.3 Additional details for the Ackley function experiment, for synthetic data in higher dimensions

The Ackley function of dimension $d$ is defined as:

$$Ackley_d : \mathcal{B} \to \mathbb{R}$$

$$x \mapsto A \exp \left( -B \sqrt{\frac{1}{d} \sum_{i=1}^{d} x_i^2} \right) + \exp \left( \frac{1}{d} \sum_{i=1}^{d} cos(cx_i) \right) - A - \exp(1)$$

where $\mathcal{B}$ is a hyperrectangle of $\mathbb{R}^d$. $(0, \dots, 0)$ is the only global optimizer of $Ackley_d$, at which the function is equal to 0. We use BoTorch's default values for $A, B, c$, which are $20, 0.2, 2\pi$ respectively.

In our experiments, we used $\mathcal{B} = [-10, 15]^d$ for all dimensions $d$.

For the TurBO baseline, we use BoTorch's default implementation, with Expected Improvement as an acquisition function, and a batch size of 1 (i.e. acquiring one point per step).

For fair comparisons, for DEUP, we use a Gaussian Process as the main model, and use its variance as the only input of the epistemic uncertainty predictor. This means that we calibrate the GP variance to match the out-of-sample squared error, using another GP to perform the regression. TurBO-DEUP is a combination of both, in which we perform the variance calibration task for the local GP models of TurBO. The uncertainty

---

[5]Plot of the function copied from https://www.sfu.ca/ ssurjano/levy13.html

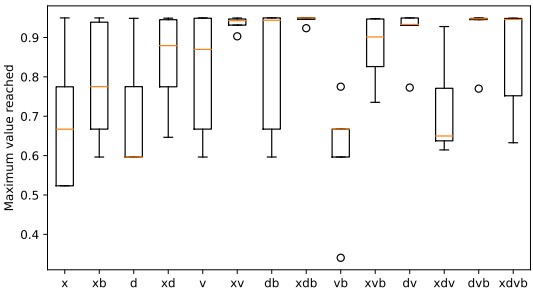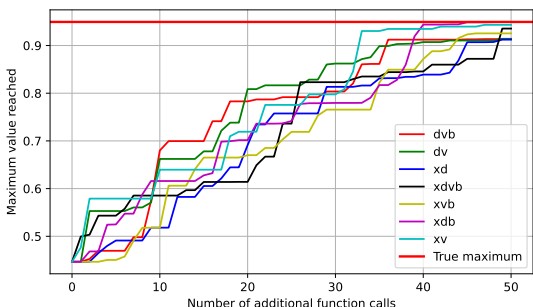

Figure 8: We train DEUP-EI to optimize the synthetic function of Fig. 3, with different subsets of the stationarizing features as inputs to the error predictor, for 50 iterations. All the instances of the algorithm start with an initial set of 6 $(x, y)$ pairs, (Left) Maximum value reached at the 40-th iteration for each subset of the stationarizing features. The box plots represent the resulting distributions across 5 different runs. (Right) Evolution of the mean (across the 5 runs) of the maximum value reached at each iteration, for the subsets of features that surpassed the 0.9 threshold.

predictor, i.e. the GP regressor, is trained with log targets, as in Appendix A.1, but also with log variances as inputs.

Note that only the stationarizing feature is used as input for the uncertainty predictor. When we used the input $x$ as well, we found that the GP error predictor overfits on the $x$ part of the input $(x, v)$, and it was detrimental to the final performances. For all experiments, we used 20 initial points.

### A.4 Ablation study for the stationarizing features

In order to study the usefulness of each subset of the "xdvb" features (respectively representing the input $x$, a density estimate, a variance estimate, and a bit indicating whether $x$ was part of the training set), we compare in Fig. 8 the different subsets of the features $\phi_z(x)$ used as input to the error predictor. Most notably, this confirms the importance of the stationarizing features besides the input $x$, and shows that incorporating $x$ to the features can help improve the performances of an interactive learner.

## B   Reinforcement Learning Experiments

For RL experiments, we used *bsuite* (Osband et al., 2020), a collection of carefully designed RL environments. *bsuite* also comes with a list of metrics which aim to evaluate RL agents from different aspects. We compare the agents based on the *basic* metric and average regret as they capture both sample complexity and final performance. The default DQN agent is used as the base of our experiments with a 3 layer fully-connected (FC) neural network as its Q-network. For the Bootstrapped DQN baseline, we used the default implementation provided by *bsuite*. To implement DQN + MC-Dropout, following the implementation from Gal & Ghahramani (2016), two dropout layers with dropout probability of 0.1 are used before the second and the third FC layers. In order take an action, the agent performs a single stochastic forward pass through the Q-network, which is equivalent to taking a sample from the posterior over the Q-values, as done in Thompson sampling, an alternative to $\epsilon-$greedy exploration.

As a density estimator, we used a Kernel Density Estimator (KDE) with a Gaussian kernel and bandwidth of 1 to map states to densities. This KDE is fit after each 10000 steps (actions) with a batch of samples from the replay buffer (which is of size 10000). The uncertainty estimator network (E-network) has the same number of layers as the Q-network, with an additional Softplus layer at the end. All other hyperparameters are the same as the default implementation by Osband et al. (2020). One complete training run for the DEUP-DQN with 5 seeds experiments takes about 0.04-0.05 GPU days on a V100 GPU. In total RL experiments took about 0.15 GPU days on a Nvidia V100 GPU.

---

**Algorithm 4** DEUP-DQN

---

Initialize replay buffer $\mathcal{D}$ with capacity $\mathcal{N}$
$Q_\theta(s, a)$: state-action value predictor
$E_\phi(\log d)$: uncertainty estimator network, which takes the log-density of the states as the input
$d(s)$: Kernel density estimator (KDE)
K: KDE fitting frequency
W: Number of warm-up episodes
**for** *episode=1 to M* **do**
    set $s_0$ as the initial state
    **for** *t=1 to max-steps-per-episode* **do**
        **with probability** $\epsilon$:   take a random action, **otherwise:**
        **if** $episode \leq$ W: $a = max_a Q_\theta(s_t, a)$, **else:** $a = max_a\big[Q_\theta(s_t, a) + \kappa \times E_\phi(\log d(s_t))(a)\big]$
        store $(s_t, a_t, r_t, s_{t+1})$ in $\mathcal{D}$
        Sample random minibatch B of transitions $(s_j, a_j, r_j, s_{j+1})$ from $\mathcal{D}$
        **if** $s_j$ is a final state: $y_j = r_j$, **else:** $y_j = r_j + \gamma max_a Q(s_t, a)$
        **Update Q-network:**
        $\theta \leftarrow \theta + \alpha_Q . \nabla_\theta \, \mathbb{E}_{(s,a)\sim B}\left[\left(y_j - Q_\theta(s, a)\right)^2\right]$
        **Update E-network:**
        $\phi \leftarrow \phi + \alpha_E . \nabla_\phi \, \mathbb{E}_{(s,a)\sim B}\left[\left[\left(y_j - Q_\theta(s, a)\right)^2 - E_\phi(\log d(s_t))(a)\right]^2\right]$
        **if** mod(*total-steps*, K) $= 0$: fit the KDE $d$ on the states of $\mathcal{D}$
    **end**
**end**

---

## C   Rejecting Difficult Examples

We adapt the standard OOD rejection task (van Amersfoort et al., 2020; Liu et al., 2020) and measure the Spearman Rank Correlation of the predicted uncertainty with the true generalization error, in addition to the OOD Detection AUROC. MC-Dropout and Deep Ensemble baselines are based on https://github.com/google/uncertainty-baselines, DUQ based on https://github.com/y0ast/deterministic-uncertainty-quantification and DUE based on https://github.com/y0ast/DUE.Note that for the ResNet50 DEUP model we continue using the ResNet-18 based DUE as variance source.

Table 2: Spearman Rank Correlation between predicted uncertainty and the true generalization error on OOD data (SVHN) with ResNet-50 models (3 seeds) trained on CIFAR-10.

| Model | ResNet-50 |
|---|---|
| MC-Dropout | $0.312 \pm 0.003$ |
| Deep Ensemble | $0.401 \pm 0.004$ |
| DUQ | $0.399 \pm 0.003$ |
| DEUP (D+V) | $\mathbf{0.465 \pm 0.002}$ |

**Training**   The baselines were trained with the CIFAR-10 training set with 10% set aside as a validation set for hyperparameter tuning. The hyperparameters are presented in Table 3 and Table 4. The hyperparameters not specified are set to the default values. For DEUP, we consider the log-density, model-variance estimate and the seen-unseen bit as the features for the error predictor. The density estimator we use is Masked-Autoregressive Flows (Papamakarios et al., 2017) and the variance estimator used is DUE (van Amersfoort et al., 2021). Note that as indicated earlier $x$, the input image, is not used as a feature for the error predictor. We present those ablations in the next sub-section. For training DEUP, the CIFAR-10 training set is divided into 5 folds, with each fold containing 8 unique classes. For each fold, we train an instance of the main predictor, density estimator and model variance estimator on only the corresponding 8 classes. The remaining

2 classes act as the out-of-distribution examples for training the error predictor. Using these folds we construct a dataset for training the error predictor, a simple feed forward network. The error predictor is trained with the log targets (i.e. log MSE between predicted and observed error). This helps since the scale of the errors varies over multiple orders of magnitude. We then train the main predictor, density estimator and the variance estimator on the entire CIFAR-10 dataset, for evaluation. The hyperparameters are presented in Table 4. For all models, we train the main predictor for 75 and 125 epochs for ResNet-18 and ResNet-50 respectively. We use SGD with Momentum (set to 0.9), with a multi-step learning schedule with a decay of 0.2 at epochs $[25, 50]$ and $[45, 90]$ for ResNet-18 and ResNet-50 respectively. One complete training run for DEUP takes about 1.5-2 GPU days on a V100 GPU. In total these set of experiments took about 31 GPU days on a Nvidia V100 GPU.

Table 3: **Left**: Hyperparameters for training Deep Ensemble (Lakshminarayanan et al., 2017). **Right**: Hyperparameters for training MC-Dropout (Gal & Ghahramani, 2016).

| Parameters | Model | |
|---|---|---|
| | **ResNet-18** | **ResNet-50** |
| Number of members | 5 | 5 |
| Learning Rate | 0.05 | 0.01 |

| Parameters | Model | |
|---|---|---|
| | **ResNet-18** | **ResNet-50** |
| Number of samples | 50 | 50 |
| Dropout Rate | 0.15 | 0.1 |
| L2 Regularization Coefficient | 6e-5 | 8e-4 |
| Learning Rate | 0.05 | 0.01 |

Table 4: **Left**: Hyperparameters for training DUQ (van Amersfoort et al., 2020). **Right**: Hyperparameters for training DUE (van Amersfoort et al., 2021).

| Parameters | Model | |
|---|---|---|
| | **ResNet-18** | **ResNet-50** |
| Gradient Penalty | 0.5 | 0.65 |
| Centroid Size | 512 | 512 |
| Length scale | 0.1 | 0.2 |
| Learning Rate | 0.05 | 0.025 |

| Parameters | Model |
|---|---|
| | **ResNet-18** |
| Inducing Points | 50 |
| Kernel | RBF |
| Lipschitz Coefficient | 2 |
| BatchNorm Momentum | 0.99 |
| Learning Rate | 0.05 |
| Weight Decay | 0.0005 |

**Ablations** We also perform some ablation experiments to study the effect of each feature for the error predictor. The Spearman rank correlation coefficient between the generalization error and the variance feature, $V$, from DUE (van Amersfoort et al., 2021) alone is $37.84 \pm 0.04$, and the log-density, $D$, from MAF (Papamakarios et al., 2017) alone is $30.52 \pm 0.03$. With only the image ($x$) the SRCC is $36.58 \pm 0.16$

Table 6 presents the results for these experiments. We observe that combining all the features performs the best. Also note that using the log-density and variance as features to the error predictor we observe better performance than using them directly, indicating that the error predictor perhaps captures a better target for the epistemic uncertainty. The boolean feature ($B$) indicating seen examples, discussed in Sec. 3.2, also leads to noticeable improvments.

Table 5: Hyperparameters for training DEUP.

| Parameters | Model | |
|---|---|---|
| | **ResNet-18** | **ResNet-50** |
| Uncertainty Predictor Architecture | [1024] x 5 | [1024] x 5 |
| Uncertainty Predictor Epochs | 100 | 100 |
| Uncertainty Predictor LR | 0.01 | 0.01 |
| Main Predictor Learning Rate | 0.05 | 0.01 |

Table 6: Spearman Rank Correlation between predicted uncertainty and the true generalization error on OOD data (SVHN) with variants of DEUP with different features as input for the uncertainty predictor. $D$ indicates the log-density from MAF (Papamakarios et al., 2017), $V$ indicates variance from DUQ (van Amersfoort et al., 2020) and $B$ indicates a bit indicating if the data is seen.

| Features | Model | |
| --- | --- | --- |
| | **ResNet-18** | **ResNet-50** |
| $D+V+B$ | $\mathbf{0.426 \pm 0.009}$ | $\mathbf{0.465 \pm 0.002}$ |
| $D+V$ | $0.419 \pm 0.003$ | $0.447 \pm 0.003$ |
| $V+B$ | $0.401 \pm 0.004$ | $0.419 \pm 0.004$ |
| $D+B$ | $0.403 \pm 0.003$ | $0.421 \pm 0.002$ |
| $x$ | $0.352 \pm 0.004$ | $0.376 \pm 0.001$ |
| $x + D + V$ | $0.382 \pm 0.006$ | $0.397 \pm 0.002$ |

## C.1 Predicting Uncertainty under Distribution Shift

We also consider the task of uncertainty estimation in the setting of shifted distributions (Ovadia et al., 2019; Hendrycks & Dietterich, 2019). We evaluate the uncertainty predictions of models trained with CIFAR-10, on CIFAR-10-C (Hendrycks & Dietterich, 2019) which consists of images from CIFAR-10 distorted using 16 corruptions like gassian blur, impulse noise, among others. Fig. 9 shows that even in the shifted distribution setting, the uncertainty estimates of DEUP correlate much better with the error made by the predictor, compared to the baselines.

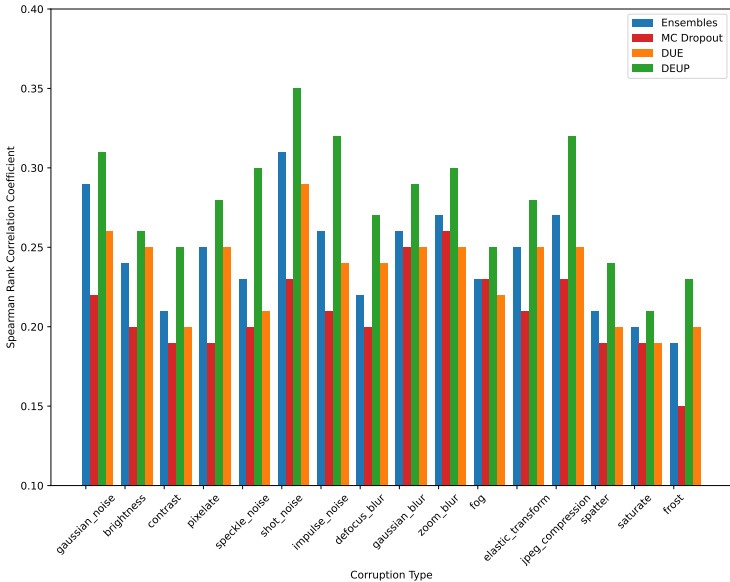

Figure 9: Spearman Rank Correlation Coefficient between the predicted uncertainty and true error for models trained with CIFAR-10, and evaluated on CIFAR-10-C. DEUP outperforms the baselines on all types of corruptions.

## D Drug Combination Experiments

To validate DEUP's uncertainty estimates in a real-world setting, we measured its performance on a regression task predicting the synergy of drug combinations. While much effort in drug discovery is spent on finding

novel small molecules, a potentially cheaper method is identifying combinations of pre-existing drugs which are synergistic (i.e., work well together). Indeed, drug combinations are the current standard-of-care for a number of diseases including HIV, tuberculosis, and some cancers (Cihlar & Fordyce, 2016; Organization & Initiative, 2010; Mokhtari et al., 2017).

However, due to the combinatorial nature of drug combinations, identifying pairs exhibiting synergism is challenging. Compounding this problem is the high monetary cost of running experiments on promising drug combinations, as well as the length of time the experiments take to complete. Uncertainty models could be used by practitioners downstream to help accelerate drug combination treatment discoveries and reduce involved development costs.

To test DEUP's performance on this task we used the DrugComb and LINCS L1000 datasets (Zagidullin et al., 2019; Subramanian et al., 2017). DrugComb is a dataset consisting of pairwise combinations of anti-cancer compounds tested on various cancer cell lines. For each combination, the dataset provides access to several synergy scores, each indicating whether the two drugs have a synergistic or antagonistic effect on cancerous cell death. LINCS L1000 contains differential gene expression profiles for various cell lines and drugs. Differential gene expressions measure the difference in the amount of mRNA related to a set of influential genes before and after the application of a drug. Because of this, gene expressions are a powerful indicator of the effect of a single drug at the cellular level.

In our experiments, each drug is represented by its Morgan fingerprint (Morgan, 1965)[6] (with 1,024 bits and a radius of 3) as well as two differential gene expression profiles (each of dimension 978) from two cell lines (PC-3 and MCF-7). In order to use gene expression features for every drug, we only used drug pairs in DrugComb where both drugs had differential gene expression data for cell lines PC-3 and MCF-7.

We first compared the quality of DEUP's uncertainty estimations to other uncertainty estimation methods on the task of predicting the combination sensitivity score (Malyutina et al., 2019) for drug pairs tested on the cell line PC-3 (1,385 examples). We evaluated the uncertainty methods using a train, validation, test split of 40%, 30%, and 30%, respectively. The underlying model used by each uncertainty estimation method consisted of a *single drug* fully connected neural network (2 layers with 2048 hidden units and output of dimension 1024) and a *combined drug* fully connected neural network (2 layers, with 128 hidden units). The embeddings of an input drug pair's drugs produced by the *single drug* network are summed and passed to the *combined drug* network, which then predicts final synergy. By summing the embeddings produced by the *single drug* network, we ensure that the model is invariant to permutations in order of the two drugs in the pair. The models were trained with Adam (Kingma & Ba, 2015), using a learning rate of 1e-4 and weight decay of 1e-5. For MC-Dropout we used a dropout probability of 0.1 on the two layers of the *combined drug* network and 3 test-time forward passes to compute uncertainty estimates. The ensemble used 3 constituent models for its uncertainty estimates. Both Ensemble and MC-Dropout models were trained with the *MSE* loss.

We also compared against DUE (van Amersfoort et al., 2021) which combines a neural network feature extractor with an approximate Gaussian process. Spectral normalization was added to all the layers of the *combined drug* network and of the *single drug* network. Let $d_{\text{emb}}$ denote the dimension of the output of the *combined drug* network, which is also the input dimension of the approximate Gaussian process. We conducted a grid-search over different values of $d_{\text{emb}}$ (from 2 to 100), the number of *inducing points* (from 3 to 200), the learning rate, and the kernel used by the Gaussian process. The highest correlation of uncertainty estimates with residuals was attained with $d_{\text{emb}} = 10$, 100 *inducing points*, a learning rate of 1e-2, and the *Matern12* kernel.

---

[6]The Morgan fingerprint represents a molecule by associating with it a boolean vector specifying its chemical structure. Morgan fingerprints have been used as a signal of various molecular characteristics to great success (Ballester & Mitchell, 2010; Zhang et al., 2006).

---

**Algorithm 5** DEUP for Drug Combinations

---

**Data:** $\mathcal{D}$ dataset of pairwise drug combinations, along with synergy scores $((d_1, d_2), y)$

**Initialization:**

Split training set into two halves, *in-sample* $\mathcal{D}_{in}$ and *out-of-sample* $\mathcal{D}_{out}$

$f_\mu(d_1, d_2)$: $\hat{\mu}$ predictor which takes a pair of drugs as input

$f_\sigma^{in}(d_1, d_2)$: In-sample $\hat{\sigma}_{in}$ error predictor

$f_\sigma^{out}(d_1, d_2)$: Out-of-sample $\hat{\sigma}_{out}$ error predictor

**Training:**

**while** *training not finished* **do**

    *In-sample update*

    Get an *in-sample* batch $(d_{1,in}, d_{2,in}, y_{in}) \sim \mathcal{D}_{in}$

    Predict $\hat{\mu} = f_\mu(d_{1,in}, d_{2,in})$ and *in-sample* error $\hat{\sigma}_{in} = f_\sigma^{in}(d_{1,in}, d_{2,in})$

    Compute *NLL*: $\frac{\log(\hat{\sigma}_{in}^2)}{2} + \frac{(\hat{\mu} - y_{in})^2}{2\hat{\sigma}_{in}^2}$

    Backpropagate through $f_\mu$ and $f_\sigma^{in}$ and update.

    *Out-of-sample update*

    Get an *out-of-sample* batch $(d_{1,out}, d_{2,out}, y_{out}) \sim \mathcal{D}_{out}$

    Estimate $\hat{\mu} = f_\mu(d_{1,out}, d_{2,out})$ and *out-of-sample* error $\hat{\sigma}_{out} = f_\sigma^{out}(d_{1,out}, d_{2,out})$

    Compute *NLL*: $\frac{\log(\hat{\sigma}_{out}^2)}{2} + \frac{(\hat{\mu} - y_{out})^2}{2\hat{\sigma}_{out}^2}$

    Backpropagate through $f_\sigma^{out}$ and update.

**end**

---

The DEUP model we used outputs two heads $\begin{bmatrix} \hat{\mu} \\ \hat{\sigma} \end{bmatrix}$ and is trained with the *NLL* $\frac{\log(\hat{\sigma}^2)}{2} + \frac{(\hat{\mu} - y)^2}{2\hat{\sigma}^2}$ in a similar fashion as in (Lakshminarayanan et al., 2017). To obtain a predictor of the out-of-sample error, we altered our optimization procedure so that the $\mu$ and $\sigma$ heads were not backpropagated through at all times. Specifically, we first split the training set into two halves, terming the former the in-sample set $\mathcal{D}_{in}$ and the latter the out-of-sample set $\mathcal{D}_{out}$. We denote as $f_\sigma^{in}$ the in-sample error predictor and $f_\sigma^{out}$ the out-of-sample error predictor. $f_\sigma^{out}$ is used to estimate total uncertainty. Note that in this setting, $f_\sigma^{out}$ predicts the square root of the epistemic uncertainty ($\hat{\sigma}_{out}$) rather than the epistemic uncertainty itself ($\hat{\sigma}_{out}^2$).

In our experiments, an extra bit is added as input to the model in order to indicate whether a given batch is from $\mathcal{D}_{in}$ or $\mathcal{D}_{out}$. Through this, the same model is used to estimate $f_\sigma^{in}$ and $f_\sigma^{out}$ with the model estimating $f_\sigma^{in}$ when the bit indicates an example is drawn from $\mathcal{D}_{in}$ and $f_\sigma^{out}$ otherwise. When the batch is drawn from $\mathcal{D}_{in}$, both heads are trained using NLL using a single forward pass. However, when the data is drawn from $\mathcal{D}_{out}$ only the $\hat{\sigma}$ head is trained. To do this, we must still predict $\hat{\mu}$ in order to compute the NLL. But the $\hat{\mu}$ predictor $f_\mu$ must be agnostic to the difference between $\mathcal{D}_{in}$ and $\mathcal{D}_{out}$. To solve this, we perform two separate forward passes. The first pass computes $\hat{\mu}$ and sets the indicator bit to 0 so $f_\mu$ has no notion of $\mathcal{D}_{out}$, while the second pass computes $\hat{\sigma}$, setting the bit to 1 to indicate the true source of the batch. Finally, we backpropagate through the $\hat{\sigma}$ head only. The training procedure is described in Algorithm 5

We report several measures for the quality of uncertainty predictions on a separate test set in Table 7.

| Model | Corr. w. res. | U. Bound | Ratio | Log Likelihood | Coverage Probability | CI width |
|---|---|---|---|---|---|---|
| MC-Dropout | $0.14 \pm 0.07$ | $0.56 \pm 0.05$ | $0.25 \pm 0.12$ | $-20.1 \pm 6.8$ | $11.4 \pm 0.2$ | $3.1 \pm 0.1$ |
| Deep Ensemble | $0.30 \pm 0.09$ | $0.59 \pm 0.04$ | $0.50 \pm 0.13$ | $-14.3 \pm 4.7$ | $10.8 \pm 1.4$ | $3.4 \pm 0.6$ |
| DUE | $0.12 \pm 0.12$ | $0.15 \pm 0.03$ | $\mathbf{0.80 \pm 0.79}$ | $-13.0 \pm 0.52$ | $15.2 \pm 1.0$ | $3.5 \pm 0.1$ |
| DEUP | $\mathbf{0.47 \pm 0.03}$ | $0.63 \pm 0.05$ | $\mathbf{0.75 \pm 0.07}$ | $\mathbf{-3.5 \pm 0.25}$ | $\mathbf{36.1 \pm 2.5}$ | $\mathbf{13.1 \pm 0.9}$ |

Table 7: Drug combinations: quality of uncertainty estimates from different methods. *Corr. w. res.* shows correlation between model residuals and predicted uncertainties $\hat{\sigma}$. A best-case *Upper Bound* on *Corr. w. res.* is obtained from the correlation between $\hat{\sigma}$ and true samples from $\mathcal{N}(0, \hat{\sigma})$. *Ratio* is the ratio between col. 1 and 2 (larger is better). *Log-likelihood*: average over 3 seeds of per sample predictive log-likelihood. *Coverage Probability*: Percentage of test samples which are covered by the 68% confidence interval. *CI width*: width of the 86% confidence interval.

For each model, we report the per sample predictive log-likelihood, coverage probability and confidence interval width, averaged over 3 seeds.

We also computed the correlation between the residuals of the model $|\hat{\mu}(x_i) - y_i|$ and the predicted uncertainties $\hat{\sigma}(x_i)$. We noted that the different uncertainty estimation methods lead to different distributions $p(\hat{\sigma}(x))$. For example, predicted uncertainties obtained with DUE always have a similar magnitude. By contrast, DEUP yields a wide range of different predicted uncertainties.

These differences between the distributions $p(\hat{\sigma}(x))$ obtained with the different methods may have an impact on the correlation metric, possibly biasing the comparison of the different methods. In order to account for differences in the distribution $p(\hat{\sigma}(x))$ across methods, we report another metric which is the ratio between the observed correlation $Corr(|\hat{\mu}(x) - y|, \hat{\sigma}(x))$ and the maximum achievable correlation given a specific distribution $p(\hat{\sigma}(x))$.

This maximum achievable correlation (refered to as the *upper bound*) is not *per se* a comparison metric, and is estimated (given a specific $p(\hat{\sigma}(x))$) as follows: we assume that, for each example $(x_i, y_i)$, the predictive distribution of the model $\mathcal{N}(\hat{\mu}(x_i), \hat{\sigma}(x_i))$ corresponds exactly to the distribution of the target, *i.e.* $y_i \sim \mathcal{N}(\hat{\mu}(x_i), \hat{\sigma}(x_i))$. Under this assumption, the residual of the mean predictor follows a distribution $\mathcal{N}(0, \hat{\sigma}(x_i))$. We can then estimate the upper bound by computing the correlation between the predicted uncertainties $\hat{\sigma}(x_i)$ and samples from the corresponding Gaussians $\mathcal{N}(0, \hat{\sigma}(x_i))$. 5 samples were drawn from each Gaussian for our evaluation. This upper bound is reported in the Table.

Finally, we reported our comparison metric: the ratio between the correlation $Corr(|\hat{\mu}(x) - y|, \hat{\sigma}(x))$ and the upper bound. The higher the ratio is, the closer the observed correlation is to the estimated upper bound and the better the method is doing.

It is interesting to note that the upper bound is much lower for DUE compared to other methods, as its predicted uncertainties lie within a short range of values.

Predicted $\hat{\mu}$ and uncertainty estimates can be visualized in Fig. 10 for different models. MC-dropout, Ensemble and DUE consistently underestimate uncertainty, while the out-of-sample uncertainties predicted by DEUP are much more consistent with the order of magnitude of the residuals. Moreover, we observed that DUE predicted very similar uncertainties for all samples, resulting in a lower upper-bound for the correlation between residuals and predicted uncertainties compared to other methods. We observed a similar pattern when experimenting with the other kernels available in the DUE package, including the standard Gaussian kernel.

Finally, we note that in the context of drug combination experiments, aleatoric uncertainty could be estimated by having access to replicates of a given experiment (*c.f.* Sec. 3), allowing us to subtract the aleatoric part from the out-of-sample uncertainty, leaving us with the epistemic uncertainty only.

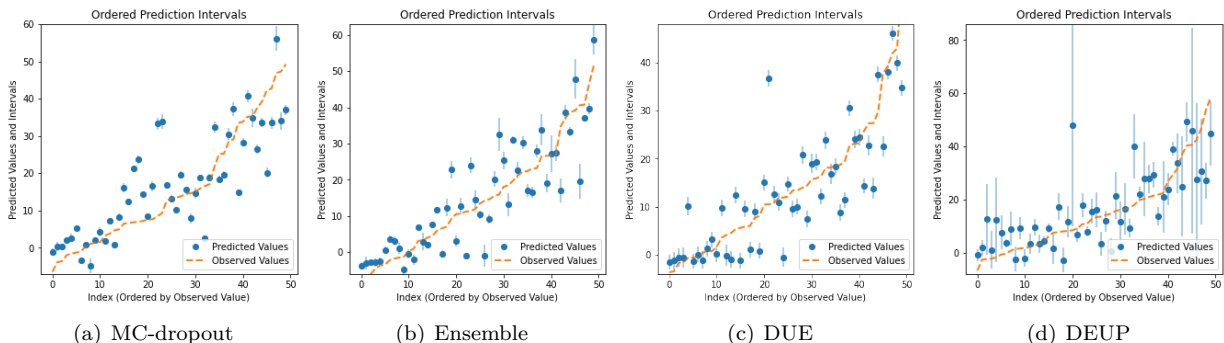

Figure 10: Predicted mean and uncertainty for different models on a separate test set. 50 examples from the test set are ordered by increasing value of true synergy score (orange). Model predictions and uncertainties are visualized in blue. MC-Dropout, Ensemble and DUE consistently underestimate the uncertainty while DEUP seems to capture the right order of magnitude. Figures made using The Uncertainty Toolbox (Chung et al., 2020).

One complete training run for the drug combination experiments takes about 0.01 GPU days on a V100 GPU. In total these set of experiments took about 0.2 GPU days on a Nvidia V100 GPU.

## E   DEUP in the presence of aleatoric uncertainty

We consider a scenario similar to that of Fig. 1, but in which Gaussian noise is added to the ground truth oracle before providing training examples. Because of the noisy training dataset, GP conflates epistemic and aleatoric uncertainty, which makes the gap between the predicted epistemic uncertainty (as measured by the GP variance) and the true epistemic uncertainty (as measured by the mean squared error between the GP mean and the noiseless ground truth function) higher than in the deterministic setting of Fig. 1. The goal of this experiment is to illustrate that using the noisy training set allows learning an estimator of aleatoric uncertainty (AU), which could be subtracted from DEUP's error predictor $e$ to obtain an estimator of epistemic uncertainty. The estimator of AU is obtained using (10). A simple linear regressor is used to estimate AU, to ovoid overfitting issues.

A key distinction of this setting, is that DEUP's training data (the errors of the main predictor) are themselves noisy, which makes it important to use more out-of-sample data to obtain reasonable total uncertainty estimates (from which we subtract the estimates of the aleatoric uncertainty).

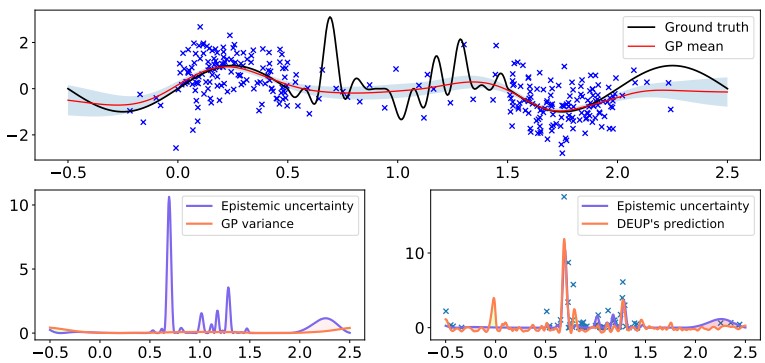

Figure 11: *Top.* A GP is trained to regress a function using noisy samples. GP uncertainty (model standard deviation) is shaded in blue. *Bottom left.* Using GP variance as a proxy for epistemic uncertainty misses out on more regions of the input space, when compared to Figure **??** . *Bottom right.* Using additional out-of-sample data in low density regions, a second GP is trained to predict the generalization error of the first GP (total uncertainty). Using second samples from the oracle for each of the training points, a linear regressor fits the training pairs $(x, \frac{1}{2}(y_1 - y_2)^2)$ to estimate the point-wise aleatoric uncertainty. Note that no constraint is imposed on DEUP's outputs, which explains the predicted negative values for uncertainties. In practice, if these predicted uncertainties were to be used, (soft) clipping should be used.

