# OpenReview forum: "DEUP: Direct Epistemic Uncertainty Prediction"
_TMLR — Accepted by TMLR_

### Review · Reviewer_TayB · 2022-11-14

**Summary Of Contributions:**

The authors of this paper argue that sticking to posterior variance as a measure of epistemic uncertainty is lacking in that it does not account for model misspecification. To follow their nomenclature, it only considers the approximation uncertainty, not the model uncertainty/bias (see Fig 2). They argue that this bias should also be modelled and create a model for it and, formulating the whole task as one of estimating the excess risk, propose to learn an error estimator which, together with a preexisting aleatoric uncertainty estimator, gives them an estimate for the excess risk. The usefulness of this excess risk estimator is then convincingly demonstrated in a series of four different experiments.


**Audience:**

Yes

**Broader Impact Concerns:**

There are none.

**Claims And Evidence:**

Yes

**Requested Changes:**

Apart from the typos mentioned in the minor section above, the main request is the change of the discussion/formulation of the model discussed at length in the weakness section. This requires a reformulation of the relevant parts to follow the common nomenclature as, e.g., summarized in the two quotes above. The simplest and, in my opinion, the best way would be to just refocus the approach on the excess risk formulation which is what the authors are relying upon anyway instead of redefining it and mixing it with terms that have other meanings.

While this might seem a minor point compared to the rest of the paper (for which I would recommend acceptance), it is an important one as the terms aleatoric and epistemic uncertainty keep getting reused and redefined seemingly at random by every other paper.


**Strengths And Weaknesses:**

The major weakness of the paper lies in the formulation of the proposed estimator as an epistemic uncertainty estimator. The method proposed by the authors serves as a convincing model for the excess risk consisting of an approximation uncertainty and a bias. Here, the former exists due to a lack of data/computational resources, while the latter exists because the model is restricted to a hypothesis space $H$ instead of an $F$ that includes the true $f^*$.
However, while a model of the uncertainty introduced due to this restriction is important, it is wrong in my opinion to formulate this as part of epistemic uncertainty, as if this epistemic uncertainty were a universal property. Rather, epistemic and aleatoric uncertainty are always conditioned on the specific hypothesis space the model lives in. To quote from the papers the authors cite in their introductory paragraph.

> "...after a long debate between us, we came to the conclusion that these concepts only make unambiguous sense if they are defined within the confines of a model of analysis. In one model an addressed uncertainty may be aleatory, in another model it may be epistemic." -- Kiureghian & Ditlevsen (2009)

> "What this example shows is that aleatoric and epistemic uncertainty should not be seen as absolute notions. Instead, they are context-dependent in the sense of depending on the setting $(X,Y,H,P)$. Changing the context will also change the sources of uncertainty: aleatoric may turn into epistemic uncertainty and vice versa. Consequently, by allowing the learner to change the setting, the distinction between these two types of uncertainty will be somewhat blurred (and their quantification will become even more difficult)." -- Hüllermeier & Waegeman (2019)

In that sense the proposed model accounts for the error, which should be properly modelled, that is due to the model being restricted to a space H, but for the model itself, this bias is not epistemic uncertainty. Rephrasing it therefore as epistemic is not correct, everything belonging to the bias is irreducibly conditioned on this space. As the quotes so nicely argue, one person's epistemic uncertainty is another one's epistemic one. In that sense, one could argue that the bias-introduced uncertainty is epistemic on the meta-level of the user applying the proposed algorithms. He/she could increase H to a space that reduces this bias up to the next level of underlying inherent aleatoric uncertainty, thereby reducing his/her epistemic uncertainty. But this applies only to this meta-level, not the current formulation of the paper.

### Minor
- Sec 2.3 "Clearly, the fact that multiple,..." -> why this clearly? If the posterior is not unimodal, but has some permutation symmetries, we do not lack information
- In (11) and later, I would suggest to change the blue color to something that overlaps less with the quotation blue
- Please follow the TMLR style guide for tables. Captions belong above the table (and below the figure)
- Table 1: DEUP (D + V) should actually include a +B following the appendix
- Typo: Table 1 column U. Bound lacks boldness
- Typo: Alg 1 in D_e specification misses a closing bracket
- Typo in Section 1: In the pitfalls paragraph SGD is used but only later introuced

---

> ### Author Response · Authors · 2022-11-15
> **Answer to reviewer**
>
> Thank you for your detailed feedback and suggestions for improvement. And thank you for bringing to our attention the typos and minor issues, which we will promptly fix.
> We would like to discuss here our different interpretations of bias and the nature of the hypothesis space that leads to the bias in order to argue for using an estimator of the excess risk as an estimator of epistemic uncertainty.
>
> > The major weakness of the paper lies in the formulation of the proposed estimator as an epistemic uncertainty estimator. The method proposed by the authors serves as a convincing model for the excess risk consisting of an approximation uncertainty and a bias. Here, the former exists due to a lack of data/computational resources, while the latter exists because the model is restricted to a hypothesis space  instead of an  that includes the true. However, while a model of the uncertainty introduced due to this restriction is important, it is wrong in my opinion to formulate this as part of epistemic uncertainty, as if this epistemic uncertainty were a universal property.
>
> We agree that in a classical parametric setting, we consider the hypothesis space $\mathcal{H}$ fixed, and in that sense the corresponding misspecification is irreducible, thus not something one would want to include as part of epistemic uncertainty. However, consider the modern setting of (typically large) deep neural networks. As more data is collected (e.g. after each round of active learning), we are allowed to choose a larger neural network (and this may be the optimal thing to do in terms of validation error) or to train the old one for longer (e.g., using validation error early stopping will allow for a larger number of training epochs because the number of training examples has increased, making the optimal effective capacity larger). Effective capacity is not only determined by the size of the network but also by the training time (more time yields greater effective capacity, lower training error). This is why we propose to quantify epistemic uncertainty in terms of excess risk, i.e., the part of the error that *can be reduced* if more data (i.e. knowledge) is acquired or more compute is available (to extract more knowledge from the data).
>
> Please let us know if you find this argument clear enough (and we should certainly do a better job of explaining it even better in the paper) or if you see any issue with it.
>
>
> > Rather, epistemic and aleatoric uncertainty are always conditioned on the specific hypothesis space the model lives in. To quote from the papers the authors cite in their introductory paragraph. (Kiureghian & Ditlevsen (2009))  -- (Hüllermeier & Waegeman (2019))
>
> While we agree with these quotes, we do not believe that our proposed method, nor our claim that an estimator of the excess risk is also an estimator of EU, contradict this point. Would it make more sense to explicitly condition our estimator of the excess risk $u(f, x)$ (Equation 9) on the hypothesis space $\mathcal{H}$ in order to make it clear that it is a sound estimator of EU ?
>
> > In that sense the proposed model accounts for the error, which should be properly modelled, that is due to the model being restricted to a space H, but for the model itself, this bias is not epistemic uncertainty.  Rephrasing it therefore as epistemic is not correct, everything belonging to the bias is irreducibly conditioned on this space. As the quotes so nicely argue, one person's epistemic uncertainty is another one's epistemic one. In that sense, one could argue that the bias-introduced uncertainty is epistemic on the meta-level of the user applying the proposed algorithms. He/she could increase H to a space that reduces this bias up to the next level of underlying inherent aleatoric uncertainty, thereby reducing his/her epistemic uncertainty. But this applies only to this meta-level, not the current formulation of the paper.
>
> Thank you for mentioning this subtle distinction between a meta-level epistemic uncertainty and the regular epistemic uncertainty. We do agree that if we need to distinguish the learner from the experimenter / machine learning practitioner, then it would be necessary to separately describe both agents' uncertainties. The main focus of our paper is learners that are able to change their hypothesis space $\mathcal{H}$ (whether explicitly or implicitly, as argued in our point above). For such learners, it is important to incorporate the bias into any measure of EU, as shown in our experiments and related works we discuss in the paper. We would like this point to be clearer in the paper, and will highlight it in the introduction and Section 2. We would also be happy to hear your suggestions on how to clarify this point.

---

> > ### Comment · Reviewer_TayB · 2022-11-16
> > **Response**
> >
> > Thank you for your quick answer. I think we are going into similar directions in our definitions.
> >
> > >  we are allowed to choose a larger neural network
> >
> > The distinction I want to draw is exactly on this _we_. The model itself is stuck with a specific architecture, i.e., space $H$. Within that it can reduce its epistemic uncertainty with more computation time or more samples. But it cannot change $H$ by itself. We are the ones who can change the architecture given the model performance and excess risk estimates, etc. and thus have our own reducible epistemic uncertainty.  Your approach seems to be a nice one for this second type, but not for the first, even though it reads in parts of the paper as if it were.
> >
> > If we formulate it of course in such a way that the model can adapt its architecture (e.g., some neural architecture search), then this becomes part of its own epistemic uncertainty again, but then we already are in a different $H'$ and not the $H$ mentioned above. (And this new $H'$ of course then potentially includes its own excess risk and so on ad infinitum...)
> >
> > > The main focus of our paper is learners that are able to change their hypothesis space (whether explicitly or implicitly, as argued in our point above).
> >
> > If we consider it in your case then I would argue that your learner actually is in the hypothesis space $F$ out of which it is able to choose subspaces $H$ and not as you define it in Definition 2 is itself in $H$ in which case it cannot know about extensions. Your experiments themselves never change $H$ (unless I overlooked something), but primarily consist in identifying new samples with the argument from my reading of the paper being that we should not rely on the epistemic uncertainty the model provides, but actually use more than that in order to get a better uncertainty estimate for ourself and to improve the model fit beyond what it could do itself. The GP in Fig 1 lower left is happy and has no (almost) epistemic uncertainty left it could reduce. DEUP then shows that actually we have some additional reducible uncertainty on a higher level which we can guide to improve the GP fit, but it won't reduce the GP's epistemic uncertainty.

---

> > > ### Author Response · Authors · 2022-11-19
> > > **Response to reviewer**
> > >
> > > Thank you for the prompt response and engaging in this important discussion.
> > >
> > > We have two main points to make:
> > >
> > > (1) Even if a human was in the loop to increase capacity (enlarge $\mathcal{H}$) when new data is collected, it would be useful to include bias in the reducible part of the error, for the following reason: in an active learning scenario with multiple rounds of acquisition, we need to choose what experiment to perform (or what example to label) which will reduce future errors as much as possible, considering that at the next round the human may increase capacity and enlarge $\mathcal{H}$ into $\mathcal{H}'$. Consider an experiment $x$ with outcome $y$, constituting later an additional example $(x,y)$ to add to the existing dataset $D$. Although the human may have chosen function class $\mathcal{H}$ by model selection when $D$ was given, it may well be that $(x,y)$ incurs a large error under that model selection (which is what the point-wise excess error is trying to estimate) and that the learner thus chooses $x$ to be queried to the oracle for the next round, yielding $D' = D \cup \{ (x,y) \}$ as a new dataset. With that new dataset, one can obtain better validation error with $\mathcal{H}'$ that has a smaller bias, and thus a smaller excess error. In this process, it is important to include bias in the calculation in order to take advantage of the possibility of reducing excess error with the right queries to the oracle.
> > >
> > >
> > > (2) Large deep neural networks trained by SGD - especially with automatic early stopping - perform a form of automatic model-selection. Given a small dataset, training will stop early to a smoother function that has smaller effective capacity. Given a larger dataset, training will continue longer and learn a more complicated function. There is indeed a bound to the largest possible capacity that the resulting effective capacity can reach, but precisely because those networks are chosen large, there is still a lot of room for varying effective capacity in practice. We thus see that we get a form of automatic model selection, with no need for a human in the loop, and this is probably the most commonly used training framework these days, thus quite important to think about and design active learning methods accordingly. In this case, there is thus no need for a human in the loop, although the result is the same: it is important to take into account the part of the excess error that can be reduced if additional data is collected and that is not just due to reduced variance but also to reduced bias. Note that something analogous is happening with Gaussian-kernel GPs if we use a validation set to pick the bandwidth of the kernel. With a small dataset the chosen bandwidth will tend to be larger than with a large dataset, and this selection can be automated.
> > >
> > > In light of these two points and the subtle distinctions you have raised, would it be clearer if in the paragraph following Eq. 8, in which we argue that an estimator the ER can be used as an estimator of the EU, if we explain that this is the case only for agents whose action space allow them to adapt their effective capacity to the dataset (because of SGD e.g.)? This would not require changing our formalism, and would entail minimal changes in the rest of the paper.

---

### Review · Reviewer_MM5Y · 2022-11-20

**Summary Of Contributions:**

The paper proposes a framework, DEUP=Direct Epistemic Uncertainty Prediction, to predict epistemic uncertainty.

The paper discusses different forms of uncertainty and looks at model misspecification as a reason why Bayesian epistemic uncertainty might not be sufficient.

Empirically, the paper shows that their proposed approach works well in several different settings and on several different tasks (Sequential Model Optimization, Reinforcement Learning, and Uncertainty Estimation for Drug Combinations, OOD Detection & Rejection Classification).

**Audience:**

Yes

**Broader Impact Concerns:**

No concerns.

**Claims And Evidence:**

No

**Requested Changes:**

Given my current state of knowledge:

## For Acceptance

1. Algorithm 1 (non-interactive) does not seem to be used in any of the experiments. Please rename it to make clear that is educational only to illustrate the original concepts. Algorithms 2 and 3 seem to be DEUP. (Otherwise, experiments for Algorithm 1 would be missing.)

2. I am not convinced that the motivation and first sections in Section 3 (DEUP) are material to why DEUP works. As such, either ablations are needed to show (11) -> (12) - which I am not sure can be done tbh - or it should be made clear that this is speculative.

    For the latter, for acceptance, I would suggest restructuring the paper such that the exposition of DEUP starts with (12) and Algorithms 2 and  3. The preceding sections are then motivation and inspiration but not theoretical grounding.

3. I would like to ask for an ablation or investigation of how (11)/(12) compares to the losses of the model trained with the new data because I don't understand how it can be a good error predictor around training data when $x$ is included in the features $\phi(x)$.

## Improvement Suggestions

1. Actual active learning experiments, given that it is mentioned a lot, and AL != SMO (or edit the paper to reduce the reader's expectations to find these).
2. Update Figure 2 according to the concerns detailed in the previous section.
3. Mention earlier in the paper that you do not require held-out/OOD data for DEUP. The initial sections make it sound like this is a necessity.
4. On page 10, there are two mentions of Algorithm 3. It looks like accidental text duplication. Please reword these.
5. Figure 6: please share the Y-axis between the two plots. Having different scaling makes it harder to compare them qualitatively.
6. How is the SRCC computed between ID and OOD data? It would seem that all OOD is equal and should have the same "rank". What is the OOD generalization error between CIFAR-10 and SVHN? (OOD is always wrong?) Please clarify this in the text if possible.
7. Appendix sections B and C and potentially others don't use `\citep` but `\cite'. Please fix this too.

I want to remind the authors that novelty and impact do not concern us reviewers for TMLR. The requested & suggested changes are about making the claims as clear and evidenced as possible. If I misunderstood or made other mistakes, my apologies, and please correct me and take this as hopefully useful signal for making edits that will help the next reader with similar experience understand the paper better.

**Strengths And Weaknesses:**

In general, I think the paper is very well written. Its empirical validation seems sound, and a preliminary code review of the supplementary material is very promising. That is, I think the code looks really good and useful and might provide a good base for additional research. In its current state, the paper will be of interest to readers in the field and provide inspiration for further research questions. However, some clarifications and changes might be necessary to either explain or make sure that all claims are backed up by evidence and put into context correctly.

The following are not necessarily weaknesses but things that might be improved via editing. Please see these as coming from a different perspective (and not having read all prior art).

## Active Learning

First, the paper mentions active learning as a setting that we can use to evaluate epistemic uncertainty:

1. It does not refer to original literature on active learning or earlier lit surveys. [Gal 2017](https://arxiv.org/abs/1703.02910) is one of the earlier papers that used EU for AL, though QBC (Query-by-Committee) from the 1990s might also count towards that depending on interpretation.
2. There are no active learning experiments after setting these expectations: SMO, which seems to me to be the non-Bayesian setting that is also dealt with by Bayesian Optimization, is not active learning by itself.
3. Algorithm 3's caption reads to be for the "active learning" setting, but it is generally used, while Algorithm 1 does not seem to be used by the paper experiments at all.

## Figure 2

Figure 2 (right) seems "wrong": H' should stay the same or become smaller with more data. If H is the universe of possible model functions, then it should stay the same. If we look at it from a Bayesian perspective (as a level set in posterior space) then it should become smaller. Imo, the important part is that $h_{z^M}$ should move closer to $h'^*$ as more data is used for training. (I still need to read Arpit 2017, but it might cause less cognitive dissonance if H' just remained as H.)

## Definition of Epistemic Uncertainty

The definition of epistemic uncertainty as approximation uncertainty + model uncertainty/bias seems to contradict the definition of epistemic uncertainty as reducible uncertainty: commonly, it is reducible by gaining access to more data, not by also changing the model. Hence, the model bias is assumed as fixed usually, is it not? Malinin et al have also decomposed (total) uncertainty into aleatoric, epistemic and model uncertainty in one of their early papers (you are probably better aware of it than I).

As such, you might want to consider that you do not actually predict epistemic uncertainty but are looking at proxies for the generalization error.

## DEUP as EU

In §3.1, the paper enumerates different examples of how to estimate aleatoric uncertainty. It would seem that in the cases where aleatoric uncertainty is set to 0, DEUP would not actually compute EU but total uncertainty or rather predict the full error. This is not epistemic uncertainty.

*I know this is much to ask (so I won't ask explicitly in the requested changes), but I will wholeheartedly ask you to consider the following request here:
Could you consider reframing the paper as Direct Reducible Error Prediction or similar? I believe the connection to that is much stronger evidenced and fits the narrative that is put forward better than claiming that it only predicts epistemic uncertainty.*

## Theory $\rightarrow$ Implementation

For me, the step from theory (11) to implemented method (12) is not clear, and a huge step: in particular, how important is $x$ in the feature vector that is used to train the error predictor?

It would seem that it should not be too important because: given that we train the predictor on the future dataset, training samples close to $x$ should be exactly where the model will have a better performance after retraining. Yet, usually, we add the most informative samples to the dataset (with the highest EU), so the error will be large for those samples, which means the error predictor will be trained to expect high losses, even though after retraining the model should have a small loss for the newly acquired data.

Are the losses updated using the newly trained models? How does this square?

## OOD via Statistical Tests

The proposed method seems related to ["A statistical framework for efficient out of distribution detection in deep neural networks"](https://arxiv.org/abs/2102.12967) and ["Density of States Estimation for Out-of-Distribution Detection"](https://arxiv.org/abs/2006.09273), in that it uses several different statistics to predict OOD. (The proposed approach except for including $x$ could be seen as predicting the error given statistics of estimates, like feature-space density or prediction variance.)

## SMO Experiment

Why do all other SMO methods perform worse than random? Given that the function is sufficiently smooth, a GP should perform better. Was the length scale chosen correctly?

---

These are a lot of questions. Answers will help me understand the paper better and come up with a recommendation.

I would like to reiterate that the paper is really well written overall. I particularly liked §2 and specifically §2.3 on model misspecification as well as the related work in §4. It's great to see these connections and explanations written down clearly.

---

> ### Author Response · Authors · 2022-11-22
> **Answer to Reviewer (1/3)**
>
> We would like to thank the reviewer for the thoughtful comments and positive view of the paper! We address your concerns below and will upload a revision with the changes based on your suggestions after the discussion.
>
> > It does not refer to original literature on active learning or earlier lit surveys. Gal 2017 is one of the earlier papers that used EU for AL, though QBC (Query-by-Committee) from the 1990s might also count towards that depending on interpretation.
>
> Thanks for mentioning the references! We will add them to the paper in Section 4.
>
> > There are no active learning experiments after setting these expectations: SMO, which seems to me to be the non-Bayesian setting that is also dealt with by Bayesian Optimization, is not active learning per-se.
>
> Thank you for raising this point! We use the term active learning to loosely refer to scenarios where a learning agent can _actively_ make decisions about what data to acquire. This can be in context of traditional active learning where we would like to select datapoints from a pool of unlabelled data to be labelled to improve the learned function, or sequential model optimization where we would like to find the maxima of a black box function. Our focus is mainly on the latter, and we will clarify this in text by referring to it as "interactive learning" to avoid confusion.
>
> > Algorithm 3's caption reads to be for the "active learning" setting, but it is generally used, while Algorithm 1 does not seem to be used by the paper experiments at all?
>
> As mentioned in the above answer, we will be changing the references to "active learning" with "interactive learning" to avoid confusion. Algorithm 1 is actually used in the OOD Detection experiments in Section 5.3.2 where we would like to learn an uncertainty estimator given a fixed dataset and use it to detect difficult OOD examples, as well as in the illustrative example of Figure 1. We will make this clear in text. Thanks for mentioning this point!
>
> > Figure 2 (right) seems "wrong": H' should stay the same or become smaller with more data. If H is the universe of possible model functions, then it should stay the same. If we look at it from a Bayesian perspective (as a level set in posterior space) then it should become smaller. Imo, the important part is that should move closer to as more data is used for training. (I still need to read Arpit 2017, but it might cause less cognitive dissonance if H' just remained as H.)
>
> It is true that the Bayesian posterior (or more generally the set of hypotheses compatible with the data) should shrink with more data. However, here H and H' represent the set of hypotheses considered before training, i.e., the prior in a Bayesian sense. When more data is available, it is a well-known result from learning theory that the optimal capacity increases with more data, i.e., a larger prior hypothesis space (H' > H) allows better generalization because we have more chances to find a function in H' closer to the ground truth f*. There is of course a trade-off (between bias and variance), but as the amount of data increases, the optimal trade-off implies greater capacity smaller bias.
> In addition, with neural networks, there is a notion of effective capacity that factors in the effect of the SGD training procedure and early stopping:
> the effective capacity (the number of randomly labeled training examples that the network can nail) increases with more training iterations, and a larger dataset generally means that early stopping will occur later, after more iterations, and thus with a larger effective capacity.
>
> > The definition of epistemic uncertainty as approximation uncertainty + model uncertainty/bias seems to contradict the definition of epistemic uncertainty as reducible uncertainty: commonly, it is reducible by gaining access to more data, not by also changing the model. Hence, the model bias is assumed as fixed usually, is it not? Malinin et al have also decomposed (total) uncertainty into aleatoric, epistemic und model uncertainty in one of their early papers (you are probably better aware of it than I). As such, you might want to consider that you do not actually predict epistemic uncertainty but are looking at proxies for the generalization error.
>
> We agree that traditional analyses of epistemic uncertainty assume the model bias to be fixed. However, in models typically used in practice like neural networks, the model bias is not fixed and is in fact reducible (as we tried to argue in Figure 2 and Section 2) as it is induced by factors such as the learning algorithms and techniques like early stopping. We believe this observation is a key contribution of our work.

---

> > ### Comment · Reviewer_MM5Y · 2022-12-12
> > **Re Figure 2**
> >
> > > > Figure 2 (right) seems "wrong": H' should stay the same or become smaller with more data. If H is the universe of possible model functions, then it should stay the same. If we look at it from a Bayesian perspective (as a level set in posterior space) then it should become smaller. Imo, the important part is that should move closer to as more data is used for training. (I still need to read Arpit 2017, but it might cause less cognitive dissonance if H' just remained as H.)
> >
> > > It is true that the Bayesian posterior (or more generally the set of hypotheses compatible with the data) should shrink with more data. However, here H and H' represent the set of hypotheses considered before training, i.e., the prior in a Bayesian sense. When more data is available, it is a well-known result from learning theory that the optimal capacity increases with more data, i.e., a larger prior hypothesis space (H' > H) allows better generalization because we have more chances to find a function in H' closer to the ground truth f*. There is of course a trade-off (between bias and variance), but as the amount of data increases, the optimal trade-off implies greater capacity smaller bias. In addition, with neural networks, there is a notion of effective capacity that factors in the effect of the SGD training procedure and early stopping: the effective capacity (the number of randomly labeled training examples that the network can nail) increases with more training iterations, and a larger dataset generally means that early stopping will occur later, after more iterations, and thus with a larger effective capacity.
> >
> > Thanks for this. Let's discuss this a bit to improve the text potentially.
> >
> > 1. You define the hypothesis space $\mathcal{H\}$ as the subset from the functions from $\mathcal{X} \to \mathcal{A}$  without taking into account the data. That is, you define it as the possible hypothesis space prior to knowing about any data. It is the *prior hypothesis space*.
> > You then use empirical risk minimization to find $h^*$.
> >
> > 2. The prior hypothesis space is data-independent by definition. There is no $\mathcal{H'}$ that changes as $M > N$.
> >
> > Hence, your arguments are not clear. You talk about the "reachable" hypothesis space in the text and refer to Figure 2, but that is not what the prior hypothesis space is as defined in the paper.
> >
> > It would seem that you at some point switch from the prior hypothesis space to a "sort-of" prior-posterior hypothesis space that is reachable through SGD and randomness (even though there might only be a few global solutions in general) given data, but that is not prior to receiving data, but it is not clear.
> >
> > More generally, is any of this really needed in the paper? Given your definition of the hypothesis space, my confusion would seem understandable, and there is no $\mathcal{H}'$ as depicted in Figure 2.
> >
> > Please update Figure 2 and clarify and simplify the text as feasible.
> >
> > Thanks!

---

> > ### Comment · Reviewer_MM5Y · 2022-12-12
> > **Re: "interactive learning" instead of "active learning"**
> >
> > > Thank you for raising this point! We use the term active learning to loosely refer to scenarios where a learning agent can actively make decisions about what data to acquire. This can be in context of traditional active learning where we would like to select datapoints from a pool of unlabelled data to be labelled to improve the learned function, or sequential model optimization where we would like to find the maxima of a black box function. Our focus is mainly on the latter, and we will clarify this in text by referring to it as "interactive learning" to avoid confusion.
> >
> > I am not sure if that is the right direction because what you describe is active learning, and in a changed revision you would have to make clear how "interactive learning" is different from "active learning" and why you don't compare to it. Also given that some prior art does use such approaches in active learning settings.
> >
> > Given the other point ((11)->(12)), this may be a lower priority, but if you want to make this paper as impactful as possible, I would highly suggest running an active learning experiment on some datasets... MNIST and CIFAR-10 or CINIC-10 maybe?

---

> ### Author Response · Authors · 2022-11-22
> **Answer to Reviewer (2/3)**
>
>
> > In §3.1, the paper enumerates different examples for how to estimate aleatoric uncertainty. It would seem that in the cases that aleatoric uncertainty is set to 0, DEUP would not actually compute EU but total uncertainty or rather predict the full error. This is not epistemic uncertainty.
>
> It is right that DEUP would predict the full error in such settings. But given the decomposition of error in Section 2, we argue that this predicted error can be decomposed into a term corresponding to aleatoric uncertainty plus a term corresponding to epistemic uncertainty, seen as the reducible error with more data and better models (which would lead to a zero error if there was no aleatoric uncertainty).
>
>
> > For me, the step from theory (11) to implemented method (12) is not clear and a huge step: in particular, how important is $x$ in the feature vector that is used to train the error predictor? It would seem that it should not be too important because: given that we train the predictor on the future dataset, training samples close to $x$ should be exactly where the model will have better performance next. Yet, usually we add the most informative samples to dataset (with highest EU), so the error will be large for those samples, which means the error predictor will be trained to expect high losses, even though after training the model should have small loss for the newly acquired data. Are the losses updated using the newly trained models? How does this square?
>
> We believe the step from (11) to (12) is actually pretty common (especially in pre-deep learning machine learning), whenever the inputs cannot be used as they are to train a predictor. $\phi_z(x)$ represents hand-designed features of the actual input pair $(x, z)$, where $z$ is a dataset. As usual, whenever we embed the inputs in a feature space, we *lose* information: the embedding with the least amount of information lost is the identity embedding (i.e. using $(x, z)$ as inputs). But there is a trade-off between the informativeness of the chosen features and their usability. Given that $x$ is usually a low-dimensional object (compared to $z$, which is a dataset of $(x, y)$ pairs), it seems natural to use $x$ as is in the feature vector.
> To "square" the difference between the error of a novel point (selected by active learning) and a point used as part of the training set, our features $\phi_z(x)$ also include a binary indicator: is x part of the training set z or not? This helps to predict the level of error expected at x.
> As confirmed by our SMO experiment, after enough training iterations, the error predictor is able to exploit these features:
>
> - If $x$ is close to inputs in the dataset $z$, then $(x, z)$ corresponds to a low error, i.e. the main predictor trained on $z$ should have a low error at x.
>
> - If $x$ is far from the inputs in $z$, then $(x, z)$ corresponds to a high error.
>
> This is the desired behaviour, and we believe it would be harder to learn it without having $x$ in the features $\phi$, unless there is a risk of overfitting to the $x$ part of $\phi$ and not using the information encoded in the other features. In the OOD detection experiments, we actually do not use $x$ in $\phi$ for this reason.
>
> We tried to explain that we are free to chose which features (including $x$) to use in $\phi$. And we would be happy to hear your suggestion as to how to make the point clearer.
>
> > The proposed method seems related to "A statistical framework for efficient out of distribution detection in deep neural networks" and "Density of States Estimation for Out-of-Distribution Detection", in that it uses several different statistics to predict OOD.
>
> Thank you for suggesting the references! They are indeed related and we will add a discussion about them in Section 4.
>
>
> > Why do all other SMO methods perform worse than random? Given that the function is sufficiently smooth, a GP should perform better. Was the length scale chosen correctly?
>
> Keep in mind that "random" means random choice of examples to label, so it is not so weak a method, especially in low-dimensional settings. This is the case only for the toy one-dimensional example in Figure 3, and not in the more interesting multi-dimensional one in Figure 4. GP-EI (or GP-UCB) tends to get stuck in local maxima (our function was specifically designed to have multiple such local maxima), and random acquisition is usually a strong baseline in 1D settings. In Figure 3, for GP experiments, the length scale is optimized at every acquisition step by maximizing the log-likelihood of the available data.

---

> > ### Comment · Reviewer_MM5Y · 2022-12-12
> > **Re: (11)->(12)**
> >
> > > > For me, the step from theory (11) to implemented method (12) is not clear and a huge step: in particular, how important is  in the feature vector that is used to train the error predictor? It would seem that it should not be too important because: given that we train the predictor on the future dataset, training samples close to  should be exactly where the model will have better performance next. Yet, usually we add the most informative samples to dataset (with highest EU), so the error will be large for those samples, which means the error predictor will be trained to expect high losses, even though after training the model should have small loss for the newly acquired data. Are the losses updated using the newly trained models? How does this square?
> >
> > > We believe the step from (11) to (12) is actually pretty common (especially in pre-deep learning machine learning), whenever the inputs cannot be used as they are to train a predictor.  represents hand-designed features of the actual input pair , where  is a dataset. As usual, whenever we embed the inputs in a feature space, we lose information: the embedding with the least amount of information lost is the identity embedding (i.e. using  as inputs). But there is a trade-off between the informativeness of the chosen features and their usability. Given that  is usually a low-dimensional object (compared to , which is a dataset of  pairs), it seems natural to use  as is in the feature vector. To "square" the difference between the error of a novel point (selected by active learning) and a point used as part of the training set, our features  also include a binary indicator: is x part of the training set z or not? This helps to predict the level of error expected at x. As confirmed by our SMO experiment, after enough training iterations, the error predictor is able to exploit these features:
> >
> > > If  is close to inputs in the dataset , then  corresponds to a low error, i.e. the main predictor trained on  should have a low error at x.
> >
> > > If  is far from the inputs in , then  corresponds to a high error.
> >
> > > This is the desired behaviour, and we believe it would be harder to learn it without having  in the features , unless there is a risk of overfitting to the  part of  and not using the information encoded in the other features. In the OOD detection experiments, we actually do not use  in  for this reason.
> >
> > > We tried to explain that we are free to chose which features (including ) to use in . And we would be happy to hear your suggestion as to how to make the point clearer.
> >
> > As mentioned in my requested changes, an ablation is needed for this. Especially to motivate the claims and how this is different from the prior art (the collection of statistics papers). Novelty is not a requirement for acceptance, but the validity of claims is.
> >
> > At the very least, it would be good to see an ablation for the different subsets of the four features, e.g. an extended Table 6 from the appendix.
> >
> > In particular, if you only used $x$ as the feature for training $D_e$, the expectation would be that $D_e$ reports high loss everywhere, given the nature of how you probably select new samples. **BTW could you add an example of $\pi(.\mid h_D, ...)$ to the main paper?**
> >
> > I still don't understand why $x$ by itself is a good idea.
> >
> > If this is commonly done in pre-deep learning literature, could you add such references and citations to the paper for me to check?
> >
> > Thanks!

---

> ### Author Response · Authors · 2022-11-22
> **Answer to Reviewer (3/3)**
>
> > Actual active learning experiments, given that it is mentioned a lot, and AL != SMO (or edit the paper to reduce the reader's expectations to find these).
>
> Thank you for raising this point. As we mentioned above, we will replace the references to active learning with interactive learning to avoid confusion for the reader.
>
> > Update Figure 2 according to the concerns detailed in the previous section.
>
> Please see our response above.
>
> > Mention earlier in the paper that you do not require held-out/OOD data for DEUP. The initial sections make it sound like this is a necessity.
>
> Thank you, we will highlight this point more in the introduction.
>
> > On page 10, there are two mentions of Algorithm 3. It looks like accidental text duplication. Please reword these.
>
> Thanks for point this out, we will fix this in the revision.
>
> > How is the SRCC computed between ID and OOD data? It would seem that all OOD is equal and should have the same "rank". What is the OOD generalization error between CIFAR-10 and SVHN? (OOD is always wrong?) Please clarify this in the text if possible.
>
> By OOD generalization error between CIFAR-10 and SVHN, we refer to the error (which is the NLL loss in this case) of the classifier trained on CIFAR 10 when evaluated on examples from SVHN (OOD examples), where the "true" labels of the OOD examples correspond to the zero vector, unlike the labels of ID examples corresponding to a one-hot vector.
>
> To compute the SRCC, we compute the true errors on OOD and unseen ID examples using the predictions from the main predictor and the ground truth labels, and compute the correlation of these errors with the predicted uncertainties. So all OOD examples actually would have a different rank based on the error made by the main predictor. On a high level, we use this metrics to establish "how well do the uncertainty estimates correlate with the errors made by the predictor".
>
> > Appendix sections B and C and potentially others don't use \citep but `\cite'. Please fix this too.
>
> Thank you for pointing this out, we will fix the usage in the appendix.

---

> > ### Comment · Reviewer_MM5Y · 2022-12-12
> > **Re: SRCC**
> >
> > > > How is the SRCC computed between ID and OOD data? It would seem that all OOD is equal and should have the same "rank". What is the OOD generalization error between CIFAR-10 and SVHN? (OOD is always wrong?) Please clarify this in the text if possible.
> >
> > > By OOD generalization error between CIFAR-10 and SVHN, we refer to the error (which is the NLL loss in this case) of the classifier trained on CIFAR 10 when evaluated on examples from SVHN (OOD examples), where the "true" labels of the OOD examples correspond to the zero vector, unlike the labels of ID examples corresponding to a one-hot vector.
> >
> > > To compute the SRCC, we compute the true errors on OOD and unseen ID examples using the predictions from the main predictor and the ground truth labels, and compute the correlation of these errors with the predicted uncertainties. So all OOD examples actually would have a different rank based on the error made by the main predictor. On a high level, we use this metrics to establish "how well do the uncertainty estimates correlate with the errors made by the predictor".
> >
> > The zero vector is generally not attainable by models that output softmax predictions. The SNGP paper shows that a uniform prediction is optimal for OOD. Further, how do you compute the NLL when your target vector is the zero vector? NLL == CrossEntropy and we take it over the target distribution (which usually is one-hot).
> >
> > Could you clarify, please? Either way, this might need to be explicitly explained in the paper.
> >
> > Also for OOD detection, it would not really matter how well OOD samples' "generalization error" and predicted uncertainty correlate. It would be more interesting to see how well predicted uncertainty predicts the generlization error for **ID data** because epistemic uncertainty should also tell us something about that.

---

> ### Author Response · Authors · 2022-12-02
> **Visibility issue**
>
> Dear reviewer,
> The Action Editor kindly pointed out that the second and third part of our response were not visible to you. We apologize for what might have looked like a delayed response. The problem should be fixed now and we're looking forward to continuing the interesting discussion.

---

> > ### Comment · Reviewer_MM5Y · 2022-12-12
> > **Thank you!**
> >
> > Yes, indeed. I did not reply to 1/3 because I was waiting for the other two. I was traveling most of last week.
> >
> > Thanks!

---

> ### Author Response · Authors · 2022-12-15
> **Updated manuscript -- Response to reviewer**
>
> Thank you for your feedback !
>
> ----
>
> Regarding your questions about SRCC:
>
> Thank you for raising this point. As we state in Section 5.3.2, we consider a per-class Binary Cross Entropy rather than a standard multi-class Cross Entropy. This is following prior work from van Amersfoort et al., (2020). So it is straightforward to compute the NLL even with a zero target vector. Note that this NLL is computed only for evaluation, but in practice, we simply use the uncertainty predicted by DEUP to make the decision. We clarified this in the updated manuscript, and rewrote that section by incorporating more details that were in the appendix.
>
> We believe the correlation between the generalization error and predicted uncertainty is useful for 2 reasons:
> a) It allows us to measure how reliable the uncertainty estimates generalize out-of-distribution. This is critical in interactive decision making where we want the uncertainty estimates to be accurate on out-of-distribution points to acquire the most informative ones.
> b) The "difficulty" of all OOD examples is not the same. For example, for a model trained on images of particular species of cats and dogs, an image of a different species is less "difficult" than an image of a car.
>
> ----
>
> Regarding your questions about Figure 2:
>
> Thank you for engaging in this interesting discussion.
> Let's take the example of Deep Learning: The hypothesis space H is defined not only through the chosen architecture of the network, but also through the pre-defined training procedure (e.g. the hyperparameters of the optimizer, the stopping criterion based on the performance on a validation set chosen from the training data...). This means that even though the learning strategy is data-independent, the (reachable) hypothesis space H depends on the training data. Arpit et al. (2017) formalized this subtle distinction using the notion of “effective capacity”. As a consequence, the learner can incorporate new training examples in areas of the input space where the bias is large, in order to change the hypothesis space H to H′, which can be closer to the Bayes predictor, thus reducing the bias, and de facto the excess risk ER.
>
> We believe that a key contribution of our work is that it takes into account that the family of function can grow in size (and thus bias be reduced) as more data becomes available, in order to estimate a reducible uncertainty that includes the effect of such bias reductions.
>
> Following your recommendation, in the updated manuscript, we removed the right part of Figure 2, and added this explanation about how more data can shift the hypothesis space in Section 2.3.
>
> ----
>
> Regarding your questions about (11) -> (12):
>
> We added two sentences before Algorithm 3 providing examples of acquisition functions $\pi$.
> We also added a few sentences after Equation 12 explaining that using x alone is not a good idea, and refering to the ablation in Table 6, in addition to the new ablation study we peformed with different subsets of the features in one of the SMO experiments. This new ablation shows that having x *in addition to* some of the other features might be helpful. We also added ablations with x as a feature in Table 6.
>
> ----
>
> Regarding "interactive learning" instead of "active learning":
>
> In the updated manuscript, we removed all references to active learning, and only refer to "interactive learning". We use interactive learning as a general setting of which classical active learning, SMO and RL are particular instantiations. We believe two of our sets of experiments (SMO, RL) should serve as enough evidence for the claims we make.
>
> Thank you

---

### Review · Reviewer_iZLw · 2022-12-03

**Summary Of Contributions:**

- A pedagogical depiction of sample/population risk, aleatoric/epistemic uncertainty, and Bayesian interpretations thereof.
- "Fit the fit" procedure to estimate error adapted to in/out distribution and/or fixed/interactive data.
- Experimental validation.


**Audience:**

Yes

**Broader Impact Concerns:**

No ethical concerns are present. The paper is a theory oriented paper and with intentions to make models more robust/transparent to invalid assumptions.

**Claims And Evidence:**

Yes

**Requested Changes:**

We do not generally find "fit-the-fit" approaches to be a compelling research contribution and were not compelled to feel otherwise from this exposition. That said we'd deeply welcome being moved from this position as it is a research area of vital importance especially as ML techniques become increasingly "launched" in the real world.

Here are two examples which we have found motivating; we list them in the hope of inspiring you to invent better alternatives:

- Masegosa '20 operates under the premise that we want to use the Bayes loss for its posterior interpretation but recognize doing so introduces suboptimal posterior predictive accuracy in the presence of model misspecifcation. Ie: the main loss is desirable and we wish merely to "tilt" it toward some objective which ideally only different in the presence of model misspecification.
- The "Density of States Estimation for Out-of-Distribution Detection" work offered a gamut of SOTA empirical results demonstrating that it was adequate to not fit the error (which is as hard as the core task) but merely fit the main model's _predictions_ and without withholding main model data or access to OOD.

Additionally, I'd like to see more effort dedicated to comparing to / showing the inadequacy of standard assumption validation practices eg Bayesian model criticsm.

However our main concern remains the same: why wouldn't a systematic discrepancy a fit-the-fit uncovers not be something I'd want to always pass "upstream" to improve the main model? If error isnt iid then doesn't this simply mean training is broken, ie failed this diagnostic test? Isnt this idea precisely what Bayesian model criticsm and Masegosa's predictive spin do already?

Regarding the overall presentation: I found the exposition to be lacking in "new idea density". The prose spent several pages on background and related work and in a way which I subjectively felt was at times too short and other times too long.  Figure 2 seemed particularly egregiously verbose; I believe left and right differ only in the inlusion of a "prime" differing lengths of arrows. Here's a few nits to offer some color on my concerns about the presentation:

Regarding the abstract, the first two sentences triggered a bit of knee-jerk reaction.

Re:first abstract sentence (and _not_ a problem in Intro definition of EU nor elsewhere): I agree it is a measure but I worry this is unecessarily vague/confusing to the uninitiated since noise in the data itself (aleatory) also imbues lack of knowledge in the learner.  As an alternative, how about "Epistemic Uncertainty is a measure of the lack of knowledge of a learner which diminishes with more
evidence."

Re:second abstract sentence:
I don't think its accurate to say that parameter uncertainty is a proxy. If you happened to know the model--because someone "in the know" told you--then it would be the appropriate definition. I also think the second sentence of the abstract could be
interpreted to imply that epistemic uncertainty cannot capture model misspecification error. In fact, if we treated the model as also a random variable then in principle we'd know how much evidence there is for a particular model. That we don't is probably because of computation and not lack of principle.


--------------


"...shown to decrease and converge towards 0 as the training dataset size grows
to infinity, which is a desirable property for EU measures."
I think this statement could be improved considerably with a reference or some additional detail.
Perhaps something along the lines of "a self-evidently reasonable property of a measure of uncertainty owing to lack of training data is that it tends to zero as training data tends to infty."?

-----

I dont think its necessary to use `x\mapsto` in `x \mapsto A(x) = sigma^2(x)` since its convention to assume the relationshoip `A(x)=\sigma^2(x)` holds for all `x`. (Indeed this is the assumption made everywhere else in the paper.)

-----




**Strengths And Weaknesses:**

Strengths:
Authors took great care to:
- establish the core ideas.
- empirically validate the approach.

Weaknesses:
- In general "fit-the-fit" does not seem practically ideal as a proposed solution. If on in-data one finds the fit to be poor, then this suggests the underlying model could be improved. If one knows the definition of OOD then one could train a classifier directly to train it.
- If one's goal is OOD, ideas such as "Density of States Estimation for Out-of-Distribution Detection" are SOTA, somewhat similar, but are not referenced. (I note that the referenced work does not succomb to the criticsm of the first point because it presumes no access to OOD data nor that error can be interpretted except insomuch as a fingerprint for "typical error.")
- Perhaps I missed it, but I was disheartened to not see treatment of the model itself as a random variable and how it is a measure of uncertainty. While a GP is in principle doing this, it does so through the choice of kernel which itself could also be regarded as a rv and whose posterior variance would in principle indicate if theirs sufficient evidence for a particular belief.  Confusingly, the authors mention these ideas eg in "Pitfalls of using the Bayesian posterior to estimate uncertainty" but then fail to adequately compare to them.
- Ab advantage of DEUP over "discrepancy-based measures of EU" is claimed that it could be adapted to use additional features. I take logical issue with this claim because why would we permit the DEUP model access features withheld from the main model.

---

> ### Author Response · Authors · 2022-12-05
> **Authors response**
>
> Thank you for your detailed feedback.
>
> We would like to first address your concern about the “fit-the-fit” approach:
> - Similar to the referenced work "Density of States Estimation for Out-of-Distribution Detection” (that we were not aware of — we will add it), DEUP does not assume access to OOD data. While Figure 1 might give the impression that a validation set is necessary, this is merely a pedagogical example to introduce DEUP. Our focus is on interactive settings, where OOD data is “free”: the data obtained at a certain timestep before the main predictor is trained on it. We will emphasize in the introduction that no OOD data, nor withheld training data / validation data, is necessary in the settings of interest.
> - Unlike the referenced work, OOD detection is merely an instance of the myriad of tasks that would benefit from accurate uncertainty estimation. In particular, we evaluate DEUP on active learning settings, where high epistemic uncertainty should drive more exploration.
> - The comment "If on in-data one finds the fit to be poor, then this suggests the underlying model could be improved. If one knows the definition of OOD then one could train a classifier directly to train it.” does not necessarily apply to interactive settings. The fit can be perfect on a given dataset, but then the learner can be “surprised” by a poor fit on new examples. As is common in active learning and SMO, such a scenario should encourage the learner to investigate the reasons of the poor fit (by e.g. querying points near the bad fit region), and then improve the model so that the fit is better in that region. This corresponds to the shift from Figure 2 Left to Figure 2 Right, where different data leads to a different reachable hypothesis space. We believe this behaviour is apparent in the SMO experiments we presented for example.
>
> All in all, DEUP does not directly address any shortcoming of discrepancy based measures of EU, except for their assumption that the model uncertainty (or bias) is fixed and can thus be ignored. DEUP argues that in practice, (1) such measures are incomplete, and thus, when EU is crucial for decision making (e.g. interactive settings), it might be preferable to use the predicted excess risk as a measure of EU, given that, (2) almost by definition, **it captures both model uncertainty and approximation uncertainty**. We opted for a verbose exposition in Section 2 to actually highlight these two points (1) and (2).
>
> -----
>
> > Ab advantage of DEUP over "discrepancy-based measures of EU" is claimed that it could be adapted to use additional features. I take logical issue with this claim because why would we permit the DEUP model access features withheld from the main model.
>
> Thank you for raising this point, and we agree that a better wording might be necessary in this sentence. Indeed we could, in principle, use the extra features as inputs to the main model directly, making discrepancy-based measures of EU implicitly rely on them as well. The point we (wrongly) tried to convey by that sentence is what we tried to explain by going from Equation 11 to equation 12: unlike the main predictor (which only cares about x), excess risk is a function of both x and the training dataset, and features conveying information about both can be used as inputs of the excess risk predictor.
>
> ----
>
> > "Perhaps I missed it, but I was disheartened to not see treatment of the model itself as a random variable and how it is a measure of uncertainty.“ +  "more effort dedicated to comparing to / showing the inadequacy of standard assumption validation practices eg Bayesian model criticsm.”
>
> We agree that we do not provide a theoretical analysis of the adequacy of our proposed estimator of EU under model misspecification. We do agree that it would be a strong contribution, and recent works we referenced ("Minimum excess risk in Bayesian learning.” and "Excess risk analysis for epistemic uncertainty with application to variational inference”) take a stab at that, albeit under strong assumptions. We can highlight this future research direction in the conclusion.
>
> ----
> Thank you for your suggestions to improve the first sentence of the abstract, and sentence about EU decreasing towards 0. We will make sure to include them in the revised version.
>
> Regarding the second sentence of the abstract, we agree that when the model is well specified, it is not merely a proxy, but is an exact measure of EU. Thank you for raising this point. We can simply replace the word “proxy” with “measure”, given that the important part of that sentence is what comes after (the fact that it’s an incomplete measure).
>
> Regarding the “x \mapsto A(x)” comment, we believe that this might just be a notation convention issue. By “x \mapsto A(x)”, we mean “the function A”, as opposed to “A(x)”, which would mean “the value A(x)”. A(x) being equal to \sigma^2(x) is not merely a convention, but is a (simple) consequence of the definitions above.

---

### Author Response · Authors · 2022-12-15
**Updated manuscript**

We would like to thank the reviewers for their thorough feedback. Following the discussions and the reviewers' recommendations, we updated the manuscript, providing more details and additional references where needed, and clarifying other parts of the text. We removed the right part of Figure 2 and tried to explain H' differently in Section 2.3. We also added an ablation study in the appendix on the usefulness of the stationarizing features in the SMO experiment, and updated Table 6 with more ablations. In the updated manuscript, the modifications with respect to the initially submitted version are in red. We would be happy to address any additional questions and concerns the reviewers might have.

---

> ### Comment · Reviewer_MM5Y · 2023-01-05
> **Thank you!**
>
> First off, happy new year! I hope you have had a break and some time to relax.
>
> Secondly: thanks so much for the updates, and thank you for your patience!
>
> I have had a look at the updates, they are great!
>
> Could you add the Morningstar and Haroush references to your section §4 as well? While I agree that they are not using the motivation from this paper, saying how e.g. the 'bag' of different metrics described in §3.2 (below (12)) could be interpreted in their context, could be very helpful to readers and provide useful placement within the literature.
>
> I'm still not convinced by (11)->(12), but the ablations are helpful. As defensive writing, I would make it clearer that a choice of other features could still be investigated. As a reader, I'm not sure why those statistics were used vs others. They seem to work well given ablations, and the motivation from (11) has been made clear. Are other possible approximations of this future work? It might improve the paper by making it clear that this was a design choice and not canonical (in the sense that someone else could choose differently given the same motivation?).
>
> The other thing that causes some heavy dissonance for me is the breadth of aleatoric uncertainty estimation options that are described which negate the term epistemic uncertainty, but I think that's covered in the future work already.
>
> However, please could you add which experiments use which aleatoric uncertainty approach (you already mention §5.1 and §5.3 for 1) and 4), but the other two are missing) in §3?
>
> Thanks!

---

### Author Response · Authors · 2023-02-03
**Thank you**

We have uploaded the camera ready version of the paper.

We would like to thank the anonymous reviewers and action editor for their continued feedback and engagement with the paper, which resulted in several improvements!

---

### Decision · Action_Editors · 2023-02-01

**Recommendation:** Accept as is

**Comment:**

I want to thank the authors and reviewers for engaging in substantive discussion, despite the review for this paper taking place over the holiday period.  It seems as though the paper was improved by the process.

**Audience:**

The paper has a clear audience in TMLR.

**Claims And Evidence:**

While the reviewers had some initial concerns about the strengths of the claims in the paper, through the review process and with modifications made by the authors, we now have all three reviewers recommending acceptance and affirming that they believe the claims made are supported by clear evidence.